

# A new machine learning method for rainfall classification: temporal random tree

Kokten Ulas Birant[1,2], Bita Ghasemkhani[3], Özlem Varlıklar[1,2] and Derya Birant[1]

[1] Department of Computer Engineering, Dokuz Eylül University, Izmir, Turkey
[2] Information Technologies Research and Application Center (DEBTAM), Dokuz Eylül University, Izmir, Turkey
[3] Graduate School of Natural and Applied Sciences, Dokuz Eylül University, Izmir, Turkey

## ABSTRACT

Traditional classification algorithms usually assume that all samples in a dataset contribute equally to the training of a machine learning model, which is not always the case. In fact, samples in temporal data, such as precipitation data, may not have equal importance; more recent samples contain more accurate and useful information than earlier ones. To address this issue, the article proposes a novel method, named temporal random tree (TRT), in which recent training samples have a greater impact on the model's decision-making process. It divides the dataset into temporal segments, assigns higher weights to classifiers trained on more recent data, and employs a weighted majority voting strategy. The experiments demonstrated the effectiveness of TRT on the real-world WeatherAUS precipitation dataset, achieving an accuracy of 83.54%, which represents a 5% improvement over the traditional random tree method. Additionally, our method achieved an average improvement of 9.98% compared to state-of-the-art results in the recent literature. These findings highlight TRT's potential as a valuable method for spatiotemporal rainfall classification.

# INTRODUCTION

Precipitation is one of the most critical meteorological phenomena, with wide-ranging impacts across various fields, including energy, agriculture, the economy, transportation, outdoor sports, mining, and water resource management. Its influence extends to numerous environmental events, such as flooding, erosion, runoff, drought, and soil infiltration, all of which are directly linked to the intensity and duration of rainfall. Given the significant role precipitation plays in both natural and human systems, accurate and reliable forecasting has become a central focus of scientific research (*Foufoula-Georgiou et al., 2020*). However, precipitation forecasting presents substantial challenges due to the inherent spatiotemporal variability of rainfall distribution. The complex interactions between atmospheric, geographical, and environmental factors make it difficult to predict

Corresponding author
Derya Birant, derya@cs.deu.edu.tr

precipitation patterns with high accuracy, especially over diverse regions and timeframes. Addressing these challenges is essential for improving decision-making in disaster management, water conservation, and agricultural planning (*Cristiano, ten Veldhuis & van de Giesen, 2017*).

Machine learning (ML) models have demonstrated significant potential in rainfall prediction due to their ability to identify complex patterns in large datasets and make accurate predictions (*Oluoch, 2024*). As a branch of artificial intelligence (AI), ML models can learn directly from the data, capturing intricate relationships between variables without explicit knowledge. This flexibility makes them particularly well-suited for dealing with the nonlinear and dynamic nature of meteorological data, including precipitation (*Zhang et al., 2025*). As a result, ML-based approaches have become increasingly popular for addressing the challenges associated with forecasting rainfall, offering improved performance over conventional methods in many cases.

Recent advances in ML have further enhanced the accuracy of precipitation predictions, offering new techniques to account for both spatial and temporal variability in rainfall patterns. State-of-the-art algorithms can process vast amounts of meteorological data, enabling more reliable predictions across geographical regions and time periods. Meteorological data encompasses a wide range of variables, including humidity, temperature, atmospheric pressure, wind speed, and historical precipitation levels (*Chantry et al., 2021*). Innovations in ML have become critical tools for managing the increasing unpredictability of weather systems driven by climate change. As weather patterns become more erratic and global temperatures rise, accurate precipitation forecasting is essential for effective disaster management, water resource planning, and adaptation strategies across multiple sectors. However, despite these advancements, further improvements are still needed to effectively address the temporal dependencies and spatial variations inherent in precipitation data (*Binetti, Massarelli & Uricchio, 2024*).

Temporal data presents unique challenges in the context of rainfall classification, primarily due to the unequal importance of data points over time. In precipitation datasets, more recent samples often carry greater relevance and accuracy, as they reflect the latest atmospheric conditions and trends. This temporal variability complicates the forecasting process, as traditional classification algorithms typically treat all data points equally, regardless of their temporal significance. As a result, older samples may contribute less insights to the model (*Balram et al., 2024*). Furthermore, conventional methods often rely on predefined assumptions about data relationships, which can hinder their ability to adapt to the dynamic nature of meteorological phenomena. It becomes essential to devise strategies that effectively overcome these limitations that prioritize recent data and effectively capture the temporal dependencies inherent in precipitation prediction (*Wang et al., 2024b*).

Alongside the importance of temporal data, spatial information plays a crucial role in improving rainfall forecasts, as geographic factors affect local weather patterns and precipitation distribution. Variations in topography, land use, and proximity to water bodies create distinct microclimates that affect rainfall and other meteorological variables. For instance, mountainous regions may experience orographic lift, leading to increased

precipitation, while urban areas might see altered patterns due to the urban heat island effect. Features such as location, temperature, humidity, and atmospheric pressure are important for capturing these geographic influences in forecasting models (*Li et al., 2022*). Incorporating spatial data enhances the understanding of how these factors interact to shape precipitation patterns, enabling more reliable predictions across diverse locations. Additionally, models must consider temporal dynamics, as precipitation patterns can change rapidly over time. Integrating both spatial and temporal data will enhance forecasting capabilities in machine learning models, contributing to improved living spaces, public health outcomes, and sustainable development objectives within the realm of spatial science (*Zhang et al., 2024*).

Despite these advancements, existing approaches for rainfall classification using machine learning still face significant limitations, particularly in handling temporal aspects. For example, in rainfall prediction systems, recent weather data is more significant for forecasting the next 24-h than data from the previous month. In this context, it is crucial to focus on recent information to enhance forecasting accuracy. However, there might also be cases when former data may help the model recognize general trends that recent data fail to show. Unfortunately, as the gap between past and present data grows, general classification algorithms can fail to learn current patterns.

To develop a rainfall classification model, various machine learning techniques—such as support vector machines (*Putri, 2024*), neural networks (*Ramani et al., 2024*), k-nearest neighbors (*Nasrullah, Saedudin & Hamami, 2024*), and ensemble methods like AdaBoost (*Kumar & Swathi, 2023*)—have been explored. Recent studies have particularly focused on deep learning and hybrid models for enhanced accuracy in hydrological time-series forecasting (*Waqas & Humphries, 2024*; *Waqas et al., 2024a*, *2024b*). A detailed survey of machine learning techniques applied to precipitation forecasting, including region-specific and AI-based approaches for both classification and regression tasks, is presented in *Waqas et al. (2023)*, *Dotse et al. (2024)*, *Putra, Rosid & Handoko (2024)*, *Waqas et al. (2024c)*, *Sokol et al. (2021)*, *Samad & Choi (2020)*. Traditional forecasting techniques, such as numerical weather prediction models and statistical methods (*e.g.*, autoregressive models, moving averages), are often based on deterministic equations and simplified physical assumptions. While grounded in meteorological theory, these approaches frequently struggle to capture the nonlinear and dynamic nature of precipitation patterns, especially under rapidly changing weather conditions. As a result, their accuracy can be limited in complex or localized environments. In contrast, machine learning approaches are data-driven and excel at uncovering hidden, nonlinear relationships within historical weather data. By learning directly from patterns in both spatial and temporal dimensions, ML models can better adapt to local climate variability and offer improved performance, particularly in short-term and regional rainfall classification. Especially, tree-based models—including decision trees (*Xiang et al., 2020*), random forests (RFs) (*Prathibha et al., 2023*), and extreme gradient boosting trees (*Woo et al., 2024*)—have been applied to precipitation data. Random tree (RT), a variant of decision trees, offers some benefits such as reduced risk of overfitting and a high level of interpretability when compared to more complex models like neural networks (*Mishra & Ratha, 2016*). These characteristics make

random trees a strong candidate for rainfall classification, yet they have not been widely applied to temporal precipitation prediction. This gap presents an opportunity to extend the RT model with temporal capabilities.

This study introduces temporal random tree (TRT), a novel machine learning approach designed to adapt to the dynamic nature of precipitation patterns in time-varying systems by incorporating spatial information. This study aims to solve the problem of short-term rainfall classification by developing a model that captures both temporal dependencies and spatial variability in meteorological data. Specifically, the model is designed to predict whether it will rain tomorrow or not, based on recent and spatially distributed weather observations. In TRT, the model prioritizes recent training samples over older ones by assigning them higher weights during the learning process. This temporal weighting mechanism ensures that the model focuses on the most relevant and up-to-date information. By emphasizing recent patterns and integrating spatial context, TRT improves the model's predictive accuracy and demonstrates substantial performance gains compared to traditional baseline methods.

The problem of predicting whether it will rain tomorrow has significant real-world applications across domains such as agriculture, transportation, disaster preparedness, and urban planning. Unlike numerical weather prediction models that estimate continuous rainfall values over longer horizons, short-term binary classification models like TRT are lightweight, fast, and suitable for real-time decision-making scenarios. This makes them valuable for time-sensitive applications such as flood alerts, logistics adjustments, and smart irrigation. Additionally, the task provides a strong benchmark for time-series learning in machine learning, helping explore challenges like imbalanced data, spatiotemporal correlations, and fast-changing environmental patterns. TRT addresses these challenges with a practical and interpretable solution that complements existing forecasting systems and supports more responsive and resilient infrastructure planning.

## Contributions

The key contributions of the proposed TRT method are as follows:

 (i) Introduction of TRT: The TRT is introduced as a novel method for rainfall classification, integrating temporal dynamics for the first time in literature.

(ii) Emphasis on recent samples: TRT improves responsiveness by assigning higher weights to recent training samples, enhancing its ability to capture relevant trends in weather data.

(iii) Application to spatial and temporal datasets: TRT is designed to handle both spatial and temporal precipitation data, expanding its applicability in meteorological research.

(iv) Investigation of regional and global classifiers: The method is tested on both regional classifiers (specific locations) and a generalized classifier (across all locations), demonstrating its versatility.

TRT was applied to a comprehensive, real-world dataset (WeatherAUS), comprising 145,460 records from 49 Australian weather stations, showing its practical relevance in real-world forecasting scenarios. The main findings of the study can be summarized as follows. TRT achieved 83.54% accuracy, a 5% improvement over the traditional RT method. Moreover, regional datasets showed a 5.1% improvement, validated by a statistically significant Mann-Whitney U test ($p$-value = 0.0003422). Furthermore, TRT outperformed existing methods, showing an average improvement of 9.98% in accuracy across the same dataset, establishing it as a competitive method in the field. In addition to accuracy, TRT performed strongly across recall, precision, and F-measure, with average values above 0.82, confirming its effectiveness in predicting both rain and no-rain events.

The structure of this article is organized as follows: "Related Works" provides an overview of related work, highlighting significant advancements pertinent to this research. "Methodology" focuses on the methodology and explains the techniques employed. "Experimental Studies" details the experimental setup and reports the findings. "Results" presents the results, providing an in-depth analysis of our findings and comparisons with existing methods. Lastly, "Discussion" concludes the study by providing a summary of the key insights and proposing directions for future research.

## RELATED WORKS

This section provides an overview of recent advancements in machine learning studies for precipitation forecasting. These studies vary significantly across several dimensions, including study region, classification or regression objectives, forecast period, and horizon. Table 1 presents a comprehensive summary of these studies, categorized by reference, year, region, method, classification/regression objectives (denoted as *C* for classification and *R* for regression), data collection period, forecast horizon, data type, and performance metrics, providing a detailed and up-to-date overview of recent 5 years of literature (*Narejo et al., 2021*; *Majnooni et al., 2023*; *Zhang et al., 2020*; *Shejule & Pekkat, 2024*; *Gianoglio et al., 2023*; *Anand & Kannan, 2022*; *Lei et al., 2024*; *Hu, Yin & Guo, 2024*; *Arbabi et al., 2024*; *Ebtehaj & Bonakdari, 2024*; *Putra, Rosid & Handoko, 2024*; *Liao, Lu & Yin, 2024*; *Wang et al., 2024a*; *Liu et al., 2024*; *Saubhagya et al., 2023*; *Skarlatos et al., 2023*; *Necesito et al., 2023*; *Baljon & Sharma, 2023*; *Kumar et al., 2023*; *Kwon et al., 2024*; *Liu et al., 2022*; *Sulaiman et al., 2022*; *Simanjuntak et al., 2022*; *Chu et al., 2022*; *Papailiou et al., 2022*; *Di Nunno et al., 2022*; *Salaeh et al., 2022*; *Poornima et al., 2023*; *Choi et al., 2021*; *Shin et al., 2021*; *Yan et al., 2021*; *Bouget et al., 2021*; *Bellido-Jiménez, Gualda & García-Marín, 2021*; *Nguyen, Kim & Bae, 2021*; *Wang et al., 2021*; *Wehbe, Temimi & Adler, 2020*; *Chhetri et al., 2020*; *Wei & Chou, 2020*). This table forms the foundation for comprehending the current research landscape and motivates the introduction of the proposed TRT method as a novel solution that addresses gaps in effectively incorporating temporal dynamics in precipitation forecasting.

Recent studies on precipitation forecasting have been conducted across diverse geographical regions. Several have focused on China (*Lei et al., 2024*; *Liao, Lu & Yin, 2024*; *Wang et al., 2024a*; *Liu et al., 2024*; *Liu et al., 2022*; *Yan et al., 2021*; *Wang et al., 2021*) and

**Table 1 Summary of recent precipitation prediction studies over the past 5 years.**

| Ref | Year | Region | Methods | C | R | Period | Forecast horizon | Data type | Performance metrics |
|---|---|---|---|---|---|---|---|---|---|
| *Lei et al. (2024)* | 2024 | China | CNN, LSTM | | √ | 2000–2018 | Monthly | Daily rainfall data from NMIC | RMSE, $R^2$, CC, SIG |
| *Hu, Yin & Guo (2024)* | 2024 | France | LSTM, RNN | √ | √ | 2016–2018 | Hourly | Radar data | MSE, MAE, SSIM, CSI, HSS |
| *Arbabi et al. (2024)* | 2024 | Iran | RF, M5, SVR, GPR, KNN | | √ | 1951–2021 | Monthly | Rainfall data from meteorological stations | $R^2$, NS, RMSE, MAE |
| *Ebtehaj & Bonakdari (2024)* | 2024 | Canada | CNN, LSTM | | √ | 1994–2022 | Hourly | Meteorological data from Quebec province | $R^2$, NSE, AIC, PBIAS, NRMSE, RSR |
| *Putra, Rosid & Handoko (2024)* | 2024 | Indonesia | XGBoost | | √ | 2022 | Hourly | Rain gauges, weather radar, weather satellite data | RMSE |
| *Liao, Lu & Yin (2024)* | 2024 | China | ConvLSTM, SmaAT-UNet | √ | | 2009–2015 | Hourly | HKO-7 radar data | POD, CSI, FAR |
| *Wang et al. (2024a)* | 2024 | China | LSTM, M-P, GAMMA | | √ | 2019–2020 | Hourly | Data from Doppler weather radar, meteorological stations, OTT-Parsivel laser raindrop spectrometer | MRE, MAE, RMSE |
| *Liu et al. (2024)* | 2024 | China, India | DFFNet, CNN | √ | | 2016–2019 | Hourly | Northern Xinjiang India, FaceDetection epilepsy NATOPS PEMS-SF | Accuracy, PRE, REC, F1 |
| *Shejule & Pekkat (2024)* | 2024 | India | LSTM | | √ | 2015–2019 | Hourly | Meteorological data | RMSE, MAPE |
| *Majnooni et al. (2023)* | 2023 | USA | RF, XGBoost, SVR, MLP, KNN, LR, AdaBoost, DT | | √ | 1983–2020 | Monthly | Rain gauge data | R, $R^2$, MSE, NSE |
| *Gianoglio et al. (2023)* | 2023 | Italy | ANN | √ | | 2017–2019 | Daily | Smart rainfall system data | REC, Specificity |
| *Saubhagya et al. (2023)* | 2023 | Sri Lanka | Spatial Kriging, CNN, SVM, NB, MLP, LSTM, Logistic Regression, RF | √ | | 2015–2019 | Daily | Weather data from MDSL | Accuracy, PRE, REC, F1 |
| *Skarlatos et al. (2023)* | 2023 | Greece | Seasonal LSTM, Univariate LSTM | | √ | 2010–2020 | Yearly | Meteorological data from GAWSN | MSE |
| *Necesito et al. (2023)* | 2023 | Philippines | DWT, Univariate LSTM | | √ | 2013–2018 | Monthly | Rainfall data from ASIT | NSE, CC, KGE, IA, LMI, MAPE, RMSE, RSR |
| *Baljon & Sharma (2023)* | 2023 | Saudi Arabia | Function Fitting ANN | √ | | 1982–2011 | Monthly | Rainfall data from metrological department | Accuracy, PRE, REC, F1, specificity |
| *Kumar et al. (2023)* | 2023 | India | CatBoost, XGBoost, Lasso, Ridge, LR, LGBM | | √ | 1980–2021 | Daily | Rainfall data from WRIS | MAE, RMSE, RMSPE, $R^2$ |
| *Kwon et al. (2024)* | 2023 | Korea | ConvLSTM, U-Net | √ | √ | 2017–2021 | Minutely | Radar data | RMSE, MAE, Accuracy, PRE, REC, F1 |

| Ref | Year | Region | Methods | C | R | Period | Forecast horizon | Data type | Performance metrics |
|-----|------|--------|---------|---|---|--------|------------------|-----------|---------------------|
| *Liu et al. (2022)* | 2022 | China | ConvLSTM | | √ | 2015–2020 | Hourly | Rainfall data from NHB | CC, MSE, RMSE, CSI, FAR, POD |
| *Sulaiman et al. (2022)* | 2022 | Malaysia | RF, PCA, SVC, SVR, ANN, RVM | √ | √ | 1998–2007 | Daily | Atmospheric data and rainfall data | Accuracy, RMSE, NSE |
| *Simanjuntak et al. (2022)* | 2022 | Indonesia | Multivariate LSTM, RF | √ | √ | 2021 | Minutely | Himawari-8 and GPM IMERG meteorological data | Accuracy, MAE, RMSE |
| *Chu et al. (2022)* | 2022 | Korea | SVM, RF, XGBoost | | √ | 2007 | Hourly | Rainfall and GIS data | RMSE, MAE, RMSLE |
| *Papailiou et al. (2022)* | 2022 | Greece | ANN, MLR | | √ | 2006–2018 | Daily | Precipitation data from the NOANN network | NSE, R, RMSE |
| *Di Nunno et al. (2022)* | 2022 | Bangladesh | M5P, SVR, M5P-SVR, PSO | | √ | 1956–2013 | Monthly | Weather data from BMD | MAE, RMSE, RAE, R² |
| *Salaeh et al. (2022)* | 2022 | Thailand | M5, RF, SVR, MLP, LSTM | | √ | 2004–2018 | Monthly | Meteorological data from TMD | MAE, RMSE, R, OI |
| *Poornima et al. (2023)* | 2022 | India | LSTM | | √ | 1901–2017 | Monthly | Rainfall data from open government data | RMSE, loss, learning rate |
| *Anand & Kannan (2022)* | 2022 | India | ANN, RF | √ | | 2012–2013 | Daily | Smart rainfall system data | PRE, REC, F1 |
| *Choi et al. (2021)* | 2021 | Japan | U-Net | √ | √ | 2017–2019 | Hourly | RAIN-F+ rainfall data | MAE, PPMCC, PRE, REC, F1 |
| *Shin et al. (2021)* | 2021 | USA Korea | Regression Tree, RF | | √ | 1996–2006 2011–2019 | Minutely | 2DVD radar data | RMSE, MAE, bias, CORR, COE, 1-NE |
| *Yan et al. (2021)* | 2021 | China | TabNet, ANN, LSTM, LightGBM | | √ | 2012–2016 | Daily | Meteorological data from stations in China | KGE, MAE, RE, RMSE, MAPE |
| *Bouget et al. (2021)* | 2021 | France | U-Net | √ | | 2016–2018 | Hourly | MeteoNet rain radar and wind data | F1, bias, TS |
| *Bellido-Jiménez, Gualda & García-Marín (2021)* | 2021 | Spain | MLP, SVM, RF, LI | | √ | 2000–2021 | Daily | Precipitation data from RIAA | RMSE, MBE, R² |
| *Nguyen, Kim & Bae (2021)* | 2021 | Korea | MLR, MARS, MLP, RNN, LSTM | | √ | 2016–2020 | Minutely | MAPLE radar data | CSI, POD, PEMR, RMSE, R, RFB |
| *Wang et al. (2021)* | 2021 | China | WPD-ELM, ARIMA, BPNN | | √ | 1958–2016 | Yearly | Precipitation data from Jinsha weather station | RMSE, MAE, R, NSEC |
| *Narejo et al. (2021)* | 2021 | Italy | DBN, CNN | | √ | 2010–2015 | Hourly | Meteorological data | MSE, RMSE, R |
| *Zhang et al. (2020)* | 2020 | China | CNN, BLSTM | | √ | 2014–2016 | Daily | Rain gauge data | RMSE, CC, MBE, MAE |

(Continued)

| Ref | Year | Region | Methods | C | R | Period | Forecast horizon | Data type | Performance metrics |
|---|---|---|---|---|---|---|---|---|---|
| *Wehbe, Temimi & Adler (2020)* | 2020 | Saudi Arabia | GWR, ANN | | √ | 2015–2018 | Daily | Ground-based rainfall data from NCM | RMSE, rBIAS, POD, FAR, PCC, NSE |
| *Chhetri et al. (2020)* | 2020 | Bhutan | MLP, CNN, LSTM, GRU, BLSTM, BLSTM-GRU | | √ | 1997–2017 | Monthly | Rainfall data from NCHM | RMSE, MSE, PCC, $R^2$ |
| *Wei & Chou (2020)* | 2020 | Taiwan | DNN, MLR | | √ | 1961–2017 | Hourly | Typhoon events data | RMSE, rRMSE, MAE, rMAE, $R^2$ |

Korea (*Kwon et al., 2024*; *Chu et al., 2022*; *Shin et al., 2021*; *Nguyen, Kim & Bae, 2021*), highlighting these areas as significant centers of research. Other regions explored include India (*Liu et al., 2024*; *Kumar et al., 2023*; *Poornima et al., 2023*), Indonesia (*Putra, Rosid & Handoko, 2024*; *Simanjuntak et al., 2022*), and Saudi Arabia (*Baljon & Sharma, 2023*; *Wehbe, Temimi & Adler, 2020*). European regions such as France (*Hu, Yin & Guo, 2024*; *Bouget et al., 2021*), Greece (*Skarlatos et al., 2023*; *Papailiou et al., 2022*), and Spain (*Bellido-Jiménez, Gualda & García-Marín, 2021*) have also been well-represented in recent literature. Research in Southeast Asia, including the Philippines (*Necesito et al., 2023*), Thailand (*Salaeh et al., 2022*), and Sri Lanka (*Saubhagya et al., 2023*), contribute to the understanding of precipitation dynamics in these regions. American regions, including Canada (*Ebtehaj & Bonakdari, 2024*) and the USA (*Shin et al., 2021*), have also been addressed in recent works.

Additionally, the reviewed works employ machine learning techniques for different forecasting objectives, with some focusing on classification and others on regression tasks. Several works address classification tasks (*Lei et al., 2024*; *Kwon et al., 2024*; *Choi et al., 2021*; *Bouget et al., 2021*), while others focus on regression objectives (*Arbabi et al., 2024*; *Ebtehaj & Bonakdari, 2024*; *Putra, Rosid & Handoko, 2024*; *Chu et al., 2022*; *Skarlatos et al., 2023*; *Kumar et al., 2023*; *Liu et al., 2022*; *Papailiou et al., 2022*; *Poornima et al., 2023*; *Bellido-Jiménez, Gualda & García-Marín, 2021*; *Wehbe, Temimi & Adler, 2020*). A subset of studies (*Hu, Yin & Guo, 2024*; *Sulaiman et al., 2022*; *Chu et al., 2022*; *Salaeh et al., 2022*; *Nguyen, Kim & Bae, 2021*) incorporate both classification and regression approaches, providing a comprehensive analysis of precipitation patterns. These varied objectives underscore the dual nature of precipitation forecasting, catering to both discrete event categorization and continuous prediction.

A wide range of machine learning methods have been utilized in precipitation forecasting, from traditional techniques like support vector machine (SVM) (*Chu et al., 2022*; *Salaeh et al., 2022*), RF (*Arbabi et al., 2024*; *Putra, Rosid & Handoko, 2024*; *Simanjuntak et al., 2022*), and logistic regression (LR) (*Bellido-Jiménez, Gualda & García-Marín, 2021*; *Wei & Chou, 2020*) to more advanced deep learning models such as convolutional neural network (CNN) (*Lei et al., 2024*; *Saubhagya et al., 2023*) and long short-term memory (LSTM) networks (*Hu, Yin & Guo, 2024*; *Liao, Lu & Yin, 2024*;

*Necesito et al., 2023*; *Waqas & Humphries, 2024*; *Waqas et al., 2024a*). Hybrid models, including ensemble methods (*Saubhagya et al., 2023*; *Sulaiman et al., 2022*), attention-based neural networks (*Yan et al., 2021*), and specialized architectures like U-Net (*Choi et al., 2021*), have been employed to capture complex patterns. Comprehensive reviews covering these approaches in various geographic and climatic contexts are provided in *Waqas et al. (2023)*. The standard classification algorithms assume equal importance for all training samples, failing to account for the temporal evolution of weather patterns. Our study proposes a new temporal data classification method, which takes into consideration the time-varying nature of the problem to enhance prediction performance.

The mentioned methods can be broadly categorized into deep learning models, hybrid and specialized architectures, ensemble and tree-based methods, support vector machines, statistical and linear models, and simpler instance-based approaches. While each category offers valuable contributions, they also present notable limitations in terms of data requirements, temporal modeling capabilities, interpretability, computational cost, and generalizability. In the following, we critically examine these categories to emphasize the specific challenges they pose and to contextualize the motivation behind our proposed method. Deep learning models (*e.g.*, CNN, LSTM, GRU, bidirectional LSTM (BiLSTM), convolutional LSTM (ConvLSTM)) have shown promising results due to their ability to learn complex temporal patterns. However, they often require large datasets, are computationally intensive, suffer from long training times, and function as black-box models with limited interpretability. These models are also sensitive to hyperparameter tuning and risk of overfitting, especially in scenarios with limited or imbalanced data.

Hybrid and specialized deep learning architectures, such as U-Net variants or attention-based networks, offer enhanced performance by incorporating domain-specific structures. Nevertheless, they tend to be complex, hard to reproduce, and highly sensitive to dataset characteristics. Their deployment in real-time or low-power environments remains a significant challenge. Tree-based models provide better interpretability and lower computational cost compared to deep learning. However, they lack inherent mechanisms to model temporal sequences and may overfit if not properly tuned, especially in high-dimensional spaces. Statistical approaches like ARIMA, ridge regression, and logistic regression are efficient and interpretable but are generally limited to linear or univariate modeling. These methods struggle with capturing nonlinear dependencies and multivariate interactions, making them less effective for complex spatiotemporal forecasting tasks.

Simpler models such as K-nearest neighbors, naive Bayes, and principal component analysis (PCA) based techniques are easy to implement but often perform poorly in high-dimensional or noisy environments. They typically lack the capacity to model temporal dynamics and require extensive preprocessing or feature engineering. This landscape highlights a key research gap: while existing models excel in either temporal modeling or interpretability, few can balance both effectively. In this context, our proposed TRT method explicitly incorporates temporal structure into a tree-based learning framework. TRT offers an interpretable, computationally efficient, and easily deployable

solution for precipitation forecasting. Unlike deep learning approaches, TRT does not require large datasets or intensive tuning and avoids the black-box limitations by maintaining transparency in its decision paths. Additionally, unlike traditional statistical models, TRT captures complex temporal interactions across multiple features.

The performance of precipitation forecasting methods was validated using various metrics based on the forecasting objectives. For example, for regression models, common metrics included root mean square deviation (RMSD) (*Lei et al., 2024*; *Arbabi et al., 2024*; *Ebtehaj & Bonakdari, 2024*), mean absolute error (MAE) (*Hu, Yin & Guo, 2024*; *Liao, Lu & Yin, 2024*; *Wang et al., 2024a*), and R-squared (*Kumar et al., 2023*; *Papailiou et al., 2022*; *Bellido-Jiménez, Gualda & García-Marín, 2021*). Classification models were evaluated using metrics, including accuracy (*Baljon & Sharma, 2023*), precision (PRE), recall (REC), and F1-score (F1) (*Liu et al., 2024*; *Saubhagya et al., 2023*). Metrics like Nash-Sutcliffe efficiency (NSE) (*Arbabi et al., 2024*; *Papailiou et al., 2022*) and structural similarity index measure (SSIM) (*Hu, Yin & Guo, 2024*) were applied for specific evaluation needs too. The reviewed research investigated various forecast horizons, with hourly forecasts being the most common (*Hu, Yin & Guo, 2024*; *Ebtehaj & Bonakdari, 2024*; *Kwon et al., 2024*; *Bouget et al., 2021*). Minutely forecasts (*Kwon et al., 2024*; *Nguyen, Kim & Bae, 2021*) were explored for very short-term predictions, while monthly and yearly horizons (*Skarlatos et al., 2023*; *Chhetri et al., 2020*) were used for long-term analysis. This range of forecast horizons illustrates the different temporal scales at which precipitation can be studied.

The superiority of the RT algorithm has been consistently recognized across a wide range of applications (*Barrios & Romero, 2019*; *Kumar et al., 2023*; *Solomon, Giwa & Taziwa, 2023*; *Kumar et al., 2022*; *Fareed et al., 2022*; *Goga, Kuyoro & Goga, 2015*; *Prakash & Nguyen, 2023*; *Abana, 2019*; *Ghasemkhani et al., 2025*). In *Barrios & Romero (2019)*, RT demonstrated superior performance over several classification techniques, including random forest (RF) and J48, indicating its competitive edge in various predictive scenarios. Similarly, in the domain of civil engineering, RT achieved higher performance than methods such as support vector machines, random forest, multiple linear regression, and multivariate adaptive regression spline (*Kumar et al., 2023*). Another study (*Solomon, Giwa & Taziwa, 2023*) showed that RT outperformed artificial neural networks in predicting the fuel properties. In the daily discharge estimation task, RT again surpassed models like random forest, M5P, REPTree, and decision stump (*Kumar et al., 2022*). Beyond classification performance, RT showed its superiority over J48, decision stump, and Hoeffding tree (*Fareed et al., 2022*). In *Prakash & Nguyen (2023)*, RT outperformed artificial neural networks in predicting load-deflection of composite concrete bridges. Its robustness has also been demonstrated in different studies (*Goga, Kuyoro & Goga, 2015*; *Abana, 2019*), where it delivered reliable and accurate results. Furthermore, advanced adaptations of the model, such as the ordinal random tree with rank-oriented feature selection (ORT-ROFS), have proven effective in complex applications like road traffic accident severity prediction (*Ghasemkhani et al., 2025*). These diverse examples collectively support the use of RT in this study and justify its extension into a temporal context through the proposed TRT method.

# METHODOLOGY

## Proposed method

In rainfall prediction systems, recent data samples tend to provide more relevant and up-to-date information compared to older ones, as climate conditions evolve over time due to factors like global warming. The consideration of many past samples in temporal data may mislead the algorithm to capture up-to-date patterns, potentially leading to prediction errors. To address this issue, we propose a novel method, called TRT, which gives more weight to recent training samples, enhancing the model's ability to determine current trends. Therefore, in the proposed method, the recent training samples have more effects on the decision of the model.

Figure 1 illustrates the architecture of the proposed approach. The dataset is composed of various weather features, including temperature, evaporation, rainfall, wind speed, sunshine, pressure, humidity, and cloud cover. The dataset is divided into several partitions, each representing a different time period, from the oldest to the most recent data. From each partition, a distinct RT classifier is constructed, reflecting the notion that different temporal datasets lead to different decision trees due to their evolving statistical patterns. The method assigns weight values to each RT, with more recent data partitions receiving higher weights to reflect their importance in capturing current weather patterns. These weighted models in an ensemble approach emphasize recency while preserving valuable historical context. Specifically, the classifier built from the most recent data is assigned the highest weight, while the classifier trained on the oldest partition is given the lowest weight. This weighting system follows a square-root-based approach. When new data is provided to the model, all random trees make predictions, which are then combined using a weighted majority voting scheme. Trees trained on more recent data have a stronger influence on the final decision, ensuring that the prediction relies on the latest available information, allowing the model to adapt to changing weather patterns.

The temporal weighting strategy in this study implicitly reflects important meteorological considerations. Seasonal variability and evolving climatic trends, often driven by global warming, affect rainfall patterns over time. Incorporating recent data with greater weight ensures the model adapts to these changes, reducing the risk of outdated patterns misleading the prediction. Additionally, spatial and temporal weather features such as temperature and humidity, which are known to be influenced by seasonal cycles, are part of the dataset, allowing the model to learn contextual climate behaviors. Thus, TRT accounts for both short-term temporal dynamics and longer-term climatic variability without requiring explicit climate modeling.

By integrating a mechanism that allows for greater emphasis on recent data, TRT effectively adapts to the dynamic nature of precipitation patterns, making it ideal for real-time prediction applications. This dynamic adaptability is further reinforced through the core structure of TRT, which enhances its ability to handle non-linearity in precipitation data. The model uses the RT algorithm, known for its ability to capture non-linear relationships between input features and the target variable. This is achieved through several mechanisms. First, random feature subset selection at each split allows the

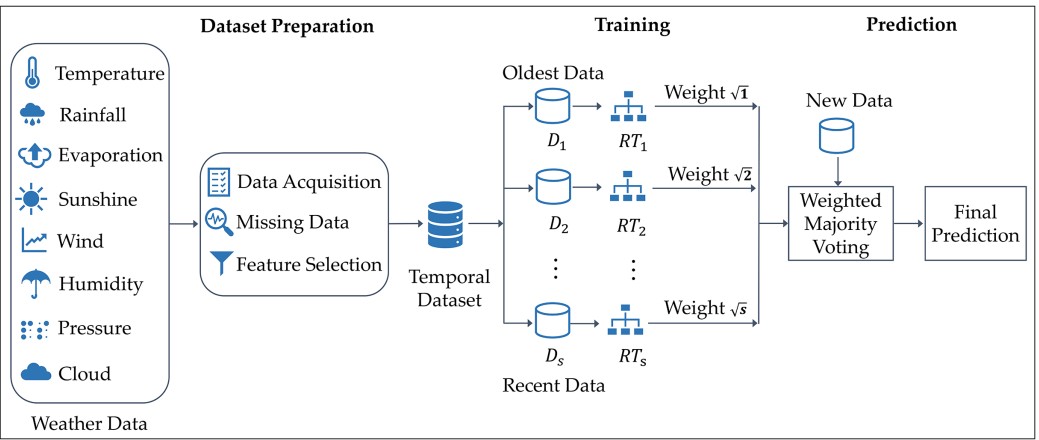

**Figure 1 Architecture of the TRT method for rainfall classification.**

model to construct more diverse and nonlinear decision boundaries. This key improvement comes from the logarithmic splitting mechanism within RT, which incorporates non-linear scaling through the K-value, aligning well with the complex, non-linear nature of real-world precipitation patterns. Second, it is different from other tree-based models in terms of the pruning issue, which preserves complex and intricate patterns in the data. Additionally, TRT reduces deterministic bias by leveraging stochasticity, encouraging the discovery of hidden interactions that may not be captured by linear methods. Furthermore, TRT introduces temporal weighting, where recent data partitions are given higher importance. This temporal framework ensures that the model adapts to evolving weather patterns, with more recent trees prioritized in the ensemble while older trees still contribute, albeit with lower weights. By emphasizing recent trends, TRT offers a more responsive model for predicting precipitation in real-time, while still retaining knowledge from past patterns to ensure a balanced and accurate prediction.

Although recurrent neural networks such as LSTM have been widely used in temporal prediction tasks, we intentionally chose not to adopt them due to several practical and methodological limitations. LSTM architectures, while powerful, are computationally demanding and often require significant hardware resources, which may not be feasible for all use cases. They involve complex tuning of hyperparameters and can be sensitive to initialization, leading to challenges in convergence. Furthermore, LSTMs act as black-box models with limited interpretability, making it difficult to extract meaningful understandings from their predictions, an important consideration in rainfall forecasting models. These challenges, combined with the need for large datasets, make LSTMs less suitable for scenarios where transparency and computational efficiency are key concerns. Our proposed TRT method was designed to overcome these issues while preserving predictive strength.

Together, these features make TRT a competitive and innovative alternative for real-time rainfall classification, particularly in the development of early warning systems.

The ability to give greater weight to recent data while maintaining a robust representation of historical patterns positions TRT as an invaluable tool for improving decision-making and response strategies in weather-related events.

## Formal description

Let the entire dataset $D$, consisting of $n$ samples, each associated with a timestamp from the set $T = \{t_1, t_2, \ldots, t_n\}$, where the timestamps are ordered such that $t_1 < t_2 \ldots < t_n$. Each timestamp $t$ corresponds to a data point $(t, x, y)$, where $x$ represents the input features, and $y$ denotes the target label. The dataset $D$ is partitioned into $s$ disjoint subsets, $D = \{D_1, D_2, \ldots, D_s\}$, according to the temporal ordering of the timestamp variable, ensuring that each subset $D_i$ covers data from a specific time range. The partitioning progresses from past to present, with $D_1$ containing the oldest data and $D_s$ containing the most recent. For each dataset $D_i$, a random tree classifier $RT_i$ is constructed. These classifiers are then assigned weights using a square-root-based weighting strategy. Specifically, the weight of the $i$th random tree is $\sqrt{i}$, meaning that recent partitions receive higher weights than older ones. The weight set can be represented as $W = \{\sqrt{1}, \sqrt{2}, \ldots, \sqrt{s}\}$.

In our weighted majority voting mechanism, the output of each classifier is assigned a weight based on its temporal proximity, and the final prediction is determined by aggregating these weighted votes, typically selecting the class with the highest total overall score. We do not normalize the weights since normalization is unnecessary for hard voting scenarios where class labels are aggregated based on weighted counts. What matters is the relative magnitude of weights between trees. Therefore, omitting normalization does not impact the correctness of the decision and aligns with the intended behavior of the voting mechanism.

## Random tree classifier

The random tree algorithm (*Frank & Kirkby, n.d.*) belongs to the family of decision tree-based methods and builds trees by injecting stochasticity into both feature and sample selection processes. This embedded randomness enhances model diversity and mitigates overfitting—benefits particularly valuable when the algorithm serves as a base learner in ensemble models. Given a training dataset $D = \{(x_i, y_i)\}_{i=1}^{n}$, where each $x_i \in R^d$ denotes a feature vector and $y_i$ is the associated target label, the goal is to explore a hypothesis space $H$ comprising all possible decision trees derivable from $D$. At each decision node, a random subset of features is selected for consideration. This subset is denoted by $F_m \subseteq F$, where $|F_m| = m$ and $m \leq d$, ensuring that only a portion of the available features is evaluated during each split. The algorithm proceeds by choosing a splitting criterion—commonly Gini impurity, entropy, or variance reduction—tailored to the specific task, such as classification or regression. These criteria aim to optimize the quality of splits by reducing impurity or uncertainty at each node. Tree construction continues recursively until a stopping condition is satisfied, which may involve a maximum allowable depth, a minimum number of data points in a node, or reaching a state where all samples in a node share the same label or exhibit negligible variance. Final predictions are derived

from the leaf nodes: classification tasks use majority voting within the leaf, while regression tasks rely on the average of target values in that node.

In classification settings, the Gini impurity is a widely used metric to quantify the impurity within a node and is defined in Eq. (1) as follows:

$$Gini(S) = 1 - \sum_{i=1}^{c} p_i^2 \tag{1}$$

where $S$ denotes the sample set at a particular node, $p_i$ is the relative frequency of class $i$ among samples in $S$, and $c$ represents the total number of distinct classes.

Entropy offers an alternative impurity metric for classification, capturing the level of unpredictability within the node. It is given by Eq. (2) as follows:

$$H(S) = - \sum_{i=1}^{c} p_i \log_2 p_i \tag{2}$$

where $p_i$ corresponds to the probability of encountering class $i$ at the node.

For regression tasks, splits are evaluated based on the goal of minimizing the variance in target values. The variance for a node $S$ is calculated using Eq. (3) as follows:

$$Var(S) = \frac{1}{|S|} \sum_{i \in S} (y_i - \bar{y})^2 \tag{3}$$

where $y_i$ indicates the target value of instance $i$, and $\bar{y}$ is the mean of the target values for all instances in node $S$.

**Temporal random tree (TRT) algorithm:** This algorithm, presented in Fig. 2, provides a detailed step-by-step procedure for implementing the proposed TRT method. It takes as input a dataset $D = \{(t_i, x_i, y_i)\}_{i=1}^{n}$, where each sample consists of timestamps $t_i$, input features $x_i$, and corresponding labels $y_i$, along with the size of the model $s$ and new query instances $Q$ for prediction. The algorithm begins with data preparation and feature selection processes. It then calculates the size of each partition by dividing the size of the dataset by the size of the model. For each $i$ from 1 to $s$, it generates partial datasets $D_i$ by the Copy function, constructs random tree classifiers $RT_i$ by the RandomTree function, and assigns weights $W_i$ to these classifiers by the Sqrt function. For classification, the algorithm iterates over each new instance $x$ and applies a weighted majority voting scheme to obtain the predicted label $y$, subsequently aggregating these predicted labels into $\widehat{Y}$. The process culminates in the output of the predicted labels for the instances in $Q$. The complexity of the TRT algorithm is $O(L(n/s) \times s)$, where $L(n/s)$ is the time required for the execution of the RT method on $n/s$ instances and $s$ is the size of the model.

## Dataset description

The dataset used in this study referred to as WeatherAUS, includes nearly 10 years of daily weather observations gathered from 49 weather stations across Australia, resulting in 145,460 records. These observations span from November 1, 2007, to June 25, 2017, providing extensive spatial and temporal coverage. The dataset is publicly and freely

```
Inputs:
    D={(t_i, x_i, y_i)}^n_{i=1}: dataset with n samples that contains features x and
                               corresponding label y on the timestamp t
    s: the size of the model
    Q: new query instances to be predicted
Output:
    Ŷ : predicted class labels for the query instances in Q
Begin:
    DataPreparation()
    FeatureSelection()
    part = |D| / s;
    for i = 1 to s
        D_i = Copy (D, part * (i − 1), part)         // Generation of partial
datasets
        RT_i = RandomTree (D_i)              // Construction of random trees
        W_i = Sqrt (i)              // Assignment of the weights
    end for
    foreach x in Q                 // Classification
        y = Voting( (RT_1, W_1), (RT_2, W_2), …, (RT_s, W_s) )
        Ŷ = Ŷ ∪ y
    end foreach
End
```

**Figure 2 Temporal random tree (TRT) algorithm.**

accessible on the Kaggle repository (*Kaggle, 2020*). The dataset includes 23 meteorological features such as temperature, rainfall, wind speed, humidity, and atmospheric pressure. These variables are used to train classification models aimed at predicting next-day rainfall. The target variable, "RainTomorrow", is a binary indicator of whether it will rain on the next day (Yes/No), while "RainToday" provides the same binary indicator for the current day.

The dataset represents a broad range of meteorological variables, as summarized in Table 2, which outlines each feature's description, units, data type, and basic statistical information (maximum, minimum, mean, and standard deviation). This table presents the full set of 23 features available in the dataset, ranging from basic weather conditions, such as MinTemp (minimum temperature) and MaxTemp (maximum temperature), to more complex variables like WindGustDir (direction of the strongest wind gust) and Pressure9am (atmospheric pressure at 9 am). WindGustDir includes several cardinal and intercardinal directions: east (E), west (W), north (N), south (S), east northeast (ENE), east southeast (ESE), west northwest (WNW), west southwest (WSW), north east (NE), north west (NW), north northeast (NNE), north northwest (NNW), south east (SE), south west (SW), south southeast (SSE), and south southwest (SSW). Additionally, it includes a binary indicator "RainToday", which is essential for predicting next-day rainfall. The feature descriptions, data types, and key statistics offer a thorough insight into the variability and range of the meteorological observations, which are crucial for building predictive models.

**Table 2 Description of the features in the rainfall classification dataset.**

| No | Feature | Description | Unit | Data type | Min | Max | Mean | Std Dev |
|---|---|---|---|---|---|---|---|---|
| 1 | Date | The date of weather observation | m/d/y | Date | 11/1/2007 | 6/25/2017 | – | – |
| 2 | Location | The meteorological station location | – | String | Adelaide, Albany, Albury, …, Woomera | | | |
| 3 | MinTemp | The lowest temperature recorded for a particular day | °C | Numeric | −8.50 | 33.90 | 12.19 | 6.40 |
| 4 | MaxTemp | The highest temperature recorded for a particular day | °C | Numeric | −4.80 | 48.10 | 23.23 | 7.12 |
| 5 | Rainfall | The amount of rainfall recorded for the day | mm | Numeric | 0 | 371.00 | 2.35 | 8.47 |
| 6 | Evaporation | The Class A pan evaporation in the 24 h until 9 am | mm | Numeric | 0 | 145.00 | 5.47 | 4.19 |
| 7 | Sunshine | The number of hours that the sun is brightly shining during the day | Hours | Numeric | 0 | 14.50 | 7.62 | 3.78 |
| 8 | WindGustDir | The direction of the strongest wind blowing in the 24 h to midnight | – | Categoric | E, W, N, S, ENE, ESE, WNW, WSW, NE, NW, NNE, NNW, SE, SW, SSE, SSW | | | |
| 9 | WindGustSpeed | The speed of the strongest wind gust in the 24 h to midnight | km/h | Numeric | 6.00 | 135.00 | 39.98 | 13.59 |
| 10 | WindDir3pm | The direction of the wind at 3 pm | – | Categoric | E, W, N, S, ENE, ESE, WNW, WSW, NE, NW, NNE, NNW, SE, SW, SSE, SSW | | | |
| 11 | WindDir9am | The direction of the wind at 9 am | – | Categoric | E, W, N, S, ENE, ESE, WNW, WSW, NE, NW, NNE, NNW, SE, SW, SSE, SSW | | | |
| 12 | WindSpeed3pm | Wind speed averaged over 10 min before 3 pm | km/h | Numeric | 0 | 87.00 | 18.64 | 8.80 |
| 13 | WindSpeed9am | Wind speed averaged over 10 min before 9 am | km/h | Numeric | 0 | 130.00 | 14.00 | 8.89 |
| 14 | Humidity3pm | The percentage of relative humidity at 3 pm | % | Numeric | 0 | 100.00 | 51.48 | 20.80 |
| 15 | Humidity9am | The percentage of relative humidity at 9 am | % | Numeric | 0 | 100.00 | 68.84 | 19.05 |
| 16 | Pressure3pm | Atmospheric pressure reduced to mean sea level at 3 pm | hpa | Numeric | 977.10 | 1,039.60 | 1,015.26 | 7.04 |
| 17 | Pressure9am | Atmospheric pressure reduced to mean sea level at 9 am | hpa | Numeric | 980.50 | 1,041.00 | 1,017.65 | 7.11 |
| 18 | Cloud3pm | Cloud-obscured fraction of the sky at 3 pm | oktas | Numeric | 0 | 9.00 | 4.50 | 2.72 |
| 19 | Cloud9am | Cloud-obscured fraction of the sky at 9 am | oktas | Numeric | 0 | 9.00 | 4.44 | 2.89 |
| 20 | Temp3pm | Observed temperature at 3 pm | °C | Numeric | −5.40 | 46.70 | 21.69 | 6.94 |
| 21 | Temp9am | Observed temperature at 9 am | °C | Numeric | −7.20 | 40.20 | 16.99 | 6.49 |
| 22 | RainToday | If the precipitation in the 24 h to 9 am exceeds 1mm, the value is "yes"; otherwise, the value is "no". | – | Binary | Yes, no | | | |
| 23 | RainTomorrow (target) | Whether it will rain tomorrow or not. | – | Binary | Yes, no (classes) | | | |

A sample dataset is presented in Table 3, showcasing observations from different spatial locations across various years. Each column in the table represents a specific sample from a unique day and weather station, illustrating the variation in meteorological conditions throughout Australia. The table provides a snapshot of essential weather features, such as temperature, rainfall, wind speed, humidity, and more. By including data from multiple locations and time points, the dataset captures the diverse climatic patterns present across the continent, offering valuable insights for training classification models aimed at predicting next-day rainfall.

**Table 3 Partial dataset showcasing daily weather observations from various stations.**

| Feature | Sample1 | Sample2 | Sample3 | Sample4 | Sample5 | Sample6 | Sample7 | Sample8 | Sample9 | Sample10 |
|---|---|---|---|---|---|---|---|---|---|---|
| Date | 11/1/2007 | 7/1/2008 | 4/22/2009 | 10/29/2010 | 5/23/2011 | 2/20/2012 | 12/23/2014 | 3/13/2015 | 1/22/2016 | 6/9/2017 |
| Location | Canberra | Melbourne | Cairns | Sydney | Melbourne | Brisbane | Hobart | Darwin | Townsville | Mildura |
| MinTemp | 8 | 9.5 | 20.2 | 14.5 | 13.4 | 21.9 | 15.7 | 22.4 | 23.7 | 3 |
| MaxTemp | 24.3 | 15.4 | 30.2 | 22.1 | 18.3 | 32.4 | 20.4 | 34.2 | 31.5 | 15.5 |
| Rainfall | 0 | 2 | 0 | 0.2 | 3.4 | 0 | 1.6 | 22.2 | 0 | 0 |
| Evaporation | 3.4 | 2.8 | 5.8 | 6.2 | 1.6 | 6 | 4.6 | 7.8 | 11 | 1 |
| Sunshine | 6.3 | 7 | 11 | 1.8 | 2 | 11.2 | 0 | 9.3 | 12.1 | 8.4 |
| WindGustDir | NW | W | SE | ENE | WSW | NE | SW | W | NE | SSE |
| WindGustSpeed | 30 | 63 | 46 | 31 | 50 | 22 | 48 | 41 | 35 | 43 |
| WindDir3pm | NW | W | ESE | ENE | SSW | NNE | S | W | ENE | S |
| WindDir9am | SW | W | S | WNW | N | W | E | WSW | ESE | SSW |
| WindSpeed3pm | 20 | 35 | 35 | 15 | 24 | 7 | 17 | 24 | 28 | 17 |
| WindSpeed9am | 6 | 37 | 20 | 11 | 13 | 4 | 4 | 22 | 17 | 11 |
| Humidity3pm | 29 | 38 | 48 | 59 | 85 | 59 | 81 | 64 | 50 | 48 |
| Humidity9am | 68 | 60 | 57 | 82 | 88 | 75 | 89 | 77 | 53 | 97 |
| Pressure3pm | 1,015 | 1,010.3 | 1,010.8 | 1,017.5 | 1,001.3 | 1,009.6 | 1,003.5 | 1,005.5 | 1,010.2 | 1,031.5 |
| Pressure9am | 1,019.7 | 1,006.8 | 1,013.8 | 1,020.7 | 1,001.2 | 1,013.9 | 1,005.3 | 1,008.8 | 1,014.2 | 1,033.7 |
| Cloud3pm | 7 | 7 | 6 | 7 | 7 | 6 | 8 | 3 | 1 | 1 |
| Cloud9am | 7 | 1 | 2 | 7 | 7 | 6 | 8 | 3 | 3 | 0 |
| Temp3pm | 23.6 | 14.6 | 28.5 | 20.5 | 13.9 | 30 | 17 | 33.8 | 30.4 | 15.3 |
| Temp9am | 14.4 | 11 | 26.3 | 16.2 | 14.5 | 26.5 | 17.4 | 30.4 | 29.1 | 5.4 |
| RainToday | No | Yes | No | No | Yes | No | Yes | Yes | No | No |
| RainTomorrow | Yes | No | No | No | Yes | Yes | Yes | No | No | No |

Table 4 presents the geographical positions, including the latitude and longitude, of the 49 weather stations included in the WeatherAUS dataset. To further visualize their spatial distribution, Fig. 3 provides a map marking each station's location across the Australian continent. This geographical representation highlights the extensive coverage of meteorological observations, showcasing how data is collected from diverse climatic zones and environments. Each station is strategically positioned to capture local weather conditions, contributing valuable insights for rainfall prediction models. The range of locations—from urban centers like Sydney and Melbourne to remote areas such as Alice Springs and Woomera—underscores the dataset's comprehensive nature, ensuring a robust analysis of weather patterns across the continent.

## Data preprocessing

In the data preprocessing phase, the target attribute, "RainTomorrow", was first addressed by removing any instances with NA (not available) values. This step ensured that the dataset contained only complete entries for the target variable, thereby enhancing the reliability of subsequent analyses. Additionally, any missing values across all attributes

**Table 4 Geographical positions of weather stations in the WeatherAUS dataset.**

| ID | Location | Latitude (S) | Longitude (E) | ID | Location | Latitude (S) | Longitude (E) |
|---|---|---|---|---|---|---|---|
| 1 | Adelaide | 34.93 | 138.60 | 26 | Nhil | 36.33 | 141.65 |
| 2 | Albany | 35.03 | 117.88 | 27 | Norah Head | 33.28 | 151.57 |
| 3 | Albury | 36.07 | 146.91 | 28 | Norfolk Island | 29.04 | 167.95 |
| 4 | Alice Springs | 23.70 | 133.88 | 29 | Nuriootpa | 34.47 | 139.00 |
| 5 | Badgerys Creek | 33.88 | 150.76 | 30 | Pearce RAAF | 31.67 | 116.02 |
| 6 | Ballarat | 37.56 | 143.85 | 31 | Penrith | 33.75 | 150.69 |
| 7 | Bendigo | 36.76 | 144.28 | 32 | Perth | 31.95 | 115.86 |
| 8 | Brisbane | 27.47 | 153.03 | 33 | Perth Airport | 31.94 | 115.97 |
| 9 | Cairns | 16.92 | 145.77 | 34 | Portland | 38.34 | 141.60 |
| 10 | Canberra | 35.28 | 149.13 | 35 | Richmond | 37.82 | 144.99 |
| 11 | Cobar | 31.50 | 145.84 | 36 | Sale | 38.11 | 147.07 |
| 12 | Coffs Harbour | 30.30 | 153.11 | 37 | Salmon Gums | 32.98 | 121.65 |
| 13 | Dartmoor | 37.92 | 141.27 | 38 | Sydney | 33.87 | 151.21 |
| 14 | Darwin | 12.46 | 130.85 | 39 | Sydney Airport | 33.95 | 151.18 |
| 15 | Gold Coast | 28.02 | 153.40 | 40 | Townsville | 19.26 | 146.82 |
| 16 | Hobart | 42.88 | 147.33 | 41 | Tuggeranong | 35.42 | 149.07 |
| 17 | Katherine | 14.45 | 132.27 | 42 | Uluru | 25.34 | 131.04 |
| 18 | Launceston | 41.43 | 147.14 | 43 | Wagga Wagga | 35.12 | 147.37 |
| 19 | Melbourne | 37.81 | 144.96 | 44 | Walpole | 34.98 | 116.73 |
| 20 | Melbourne Airport | 37.67 | 144.84 | 45 | Watsonia | 37.71 | 145.08 |
| 21 | Mildura | 34.21 | 142.13 | 46 | Williamtown | 32.81 | 151.84 |
| 22 | Moree | 29.46 | 149.84 | 47 | Witchcliffe | 34.02 | 115.10 |
| 23 | Mount Gambier | 37.83 | 140.78 | 48 | Wollongong | 34.42 | 150.89 |
| 24 | Mount Ginini | 35.53 | 148.77 | 49 | Woomera | 31.20 | 136.83 |
| 25 | Newcastle | 32.93 | 151.78 | | | | |

were replaced with a placeholder "?" to facilitate further examination and prevent data loss. The dataset was then sorted chronologically from oldest to newest, allowing for a clear temporal analysis of the weather patterns.

## Feature selection

Feature selection plays a crucial role in machine learning, aimed at reducing the dimensionality of the dataset while minimizing information loss. In this study, feature selection was performed to recognize the most relevant attributes for predicting rainfall, specifically the target variable "RainTomorrow". The focus was on selecting features that contribute significantly to model accuracy while reducing dimensionality for improved computational efficiency. To rank the features according to their importance, we utilized Information Gain as one of the primary techniques for feature selection. This method evaluates the contribution of each feature in relation to the target variable by measuring the
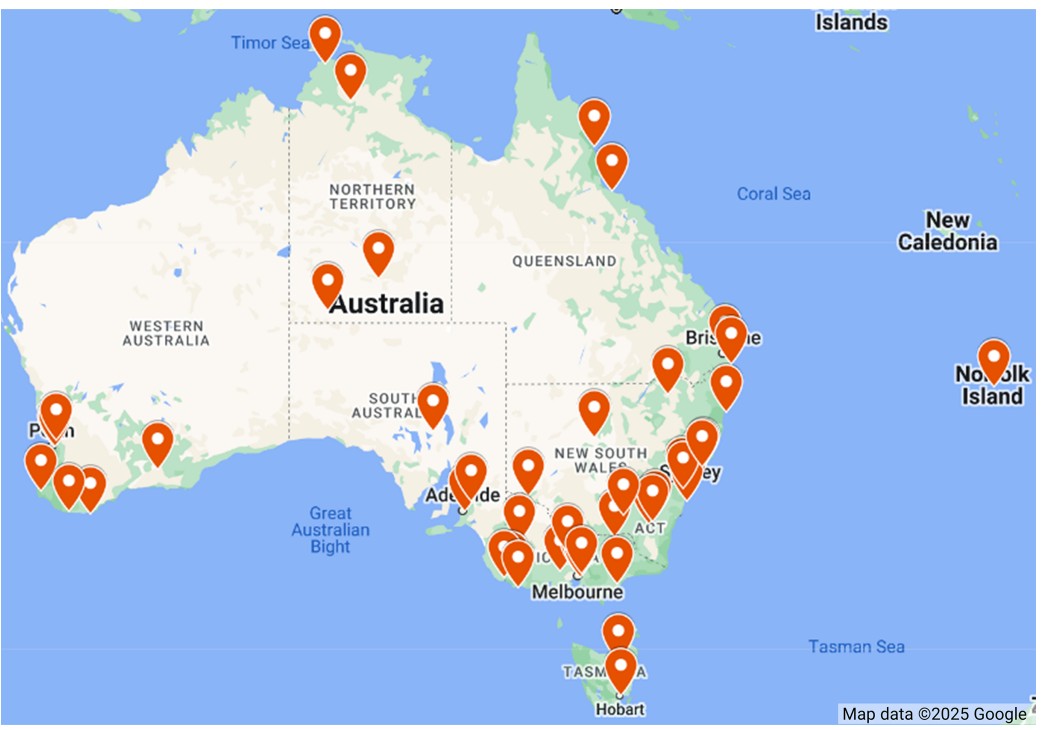

**Figure 3 Spatial distribution of weather stations in the WeatherAUS dataset, Map data ©2025 Google.**

mutual information. Specifically, it calculates the entropy on a particular attribute, as expressed in Eq. (4):

$$Information\ Gain\ (S, A) = Entropy\ (S) - \sum_{v\ \in\ values(A)} \frac{|S_v|}{|S|} Entropy\ (S_v) \tag{4}$$

where Entropy is defined by Eq. (5):

$$Entropy\ (S) = -\sum_{i=1}^{c} p_i \log_2(p_i). \tag{5}$$

In this context, $S$ represents the dataset, $A$ is an attribute, $v$ denotes all possible values of the attribute $A$, and $S_v$ is the subset of $S$ where the attribute $A$ has the value $v$. The $p_i$ represents the proportion of instances belonging to the class $i$, with $c$ denoting the overall count of classes in the target variable "RainTomorrow". For example, in this case, the classes could be "Yes" (it will rain) and "No" (it will not rain). We established a threshold of 0.01, ensuring that features with an Information Gain below this level were discarded. This approach allows for the retention of only the most informative attributes, thereby enhancing the efficiency of the model and reducing the risk of overfitting.

Figure 4 illustrates the feature importance scores across the selected features. It is evident that "Humidity3pm" and "Rainfall" exhibit the highest feature importance,
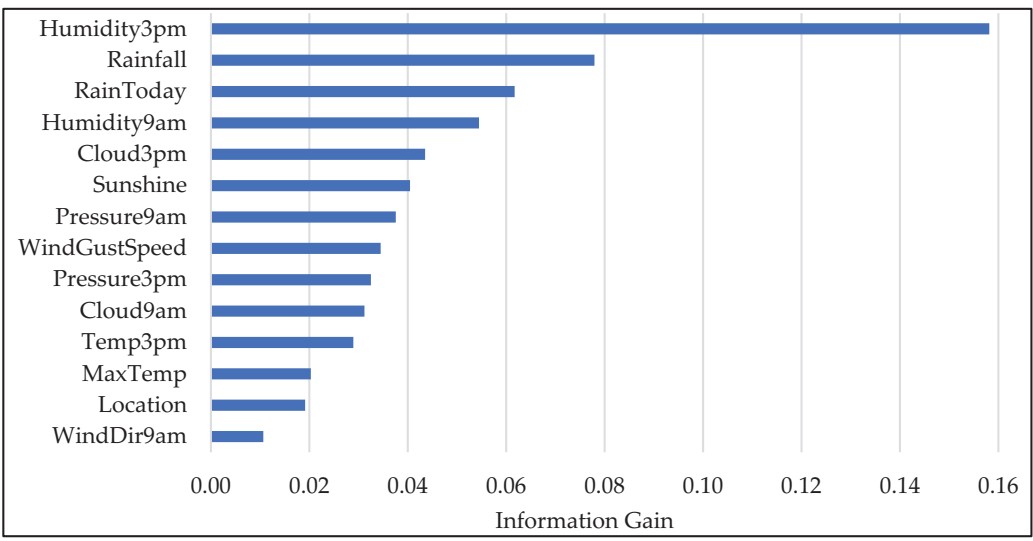

**Figure 4** The "Information Gain" values for selected features in the WeatherAUS dataset.

indicating their significant contribution to predicting the target variable, "RainTomorrow". Other prominent features include "RainToday", "Humidity9am", "Cloud3pm", and "Sunshine", which also provide valuable information. The graphical representation underscores the importance of these features in our predictive model.

In the current study, we analyzed the impact of feature removal performance. This involved systematically observing how the absence of features affected the overall accuracy and robustness of the model. Common heuristic approaches from the literature were considered for feature subset selection. These heuristics offer simple yet effective strategies to reduce dimensionality. Specifically, $\log_2(n)$ and $\sqrt{n}$ heuristics were examined, where $n$ represents the number of all features ($n = 22$ in the WeatherAUS dataset), resulting in approximate subset sizes of four or five features. Therefore, the model's performance was tested using the top-4 and top-5 ranked features. Although these reduced subsets achieved reasonable accuracy (81.13% for the top-4 and 81.64% for the top-five features), the best overall performance (83.54%) was obtained when a 0.01 threshold was used. Therefore, considering this threshold, the top-14 feature set was used in the final model to maximize predictive performance. This selection provided a good trade-off between model complexity and generalization, ultimately enhancing the classifier's ability to make accurate predictions on unseen data.

## EXPERIMENTAL STUDIES

This article presents a novel approach, called TRT, designed to emphasize more recent training data, influencing the model's decision-making more significantly. The TRT method was evaluated on datasets containing both spatial and temporal precipitation data, where it showcased its effectiveness. To demonstrate its practical application, the model

was tested on the real-world WeatherAUS dataset. The implementation of TRT was developed in C# using the Weka libraries (*Witten et al., 2016*).

Although the proposed TRT model is primarily designed for short-term rainfall classification—specifically, forecasting the likelihood of rain on the following day—the temporal weighting mechanism it employs has the potential to be beneficial for long-term prediction as well. To find the probability of rain on each day of the upcoming week, the algorithm can be applied by recursive forecasting (predict day 1, then use that to help predict day 2, and so on). By giving more weight to recent data, the model can quickly adapt to changing environmental conditions, making it more responsive to current trends. This responsiveness is crucial for forecasting tasks, especially when dealing with evolving patterns. The temporal weighting system also reduces the influence of outdated or irrelevant data, improving the model's overall performance. When applied to long-term forecasting, the temporal weighting approach it utilizes could be adapted by providing a proper balance between recent trends and historical context.

The configuration and hyperparameters used for the RT classifier in our experiments are detailed in Table 5. These parameters control various aspects of the model's behavior, such as the number of attributes randomly selected at each node as "KValue", the maximum depth of the tree "maxDepth", and the minimum number of instances per leaf "minNum". Additionally, the parameter "minVarianceProp" influences the splitting criteria for numeric attributes. The parameter "$s$" represents the number of temporal partitions into which the training data is divided, corresponding to the number of individual random trees constructed in the ensemble, as the size of the TRT model. A larger value of "$s$" increases temporal resolution by enabling more precise emphasis on recent data within the prediction process. To evaluate the sensitivity of the model's square-root-based weighting to different values of "$s$" (ranging from 2 to 10), a series of experiments were conducted, and the results are presented in Table 6. Performance is evaluated using accuracy, precision, recall, and F-measure. The results indicate that the performance consistently improves as "$s$" increases, with the best accuracy (83.54%) and F-measure (0.830) achieved at $s = 10$. This trend demonstrates that the model benefits from finer temporal granularity, reinforcing the importance of recent data in improving predictive performance. Furthermore, to empirically evaluate the effect of the proposed square-root-based weighted voting scheme on TRT model robustness, we compared it against two alternative voting strategies, including simple weighted voting and power-weighted voting. In all three strategies, the temporal order of the trees is respected— trees trained on more recent data are given greater influence in the voting process. In the simple weighted voting approach, each tree is assigned a weight directly proportional to its temporal position (*e.g.*, weight = 1 for the oldest, 2 for the next, …, up to s for the most recent tree). This linear scheme modestly favors recent data. The power-weighted voting method intensifies this bias by assigning exponential weights (*i.e.*, weight = $2^1$ for the oldest, $2^2$ for the next, …, $2^s$ for the most recent), resulting in a strong emphasis on the most recent data. Our proposed square-root-based weighted voting strategy, in contrast,

**Table 5 Hyperparameters of the classifier used in the TRT method.**

| Hyperparameter | Description | Value |
|---|---|---|
| KValue | Number of randomly chosen attributes at each node. If 0, it defaults to $\log_2(\#predictors) + 1$. | 0 |
| maxDepth | The tree's maximum depth. If 0, the tree grows until all leaves are pure or contain fewer than the minimum number of instances. | 0 |
| minNum | Minimum number of samples per leaf. | 1.0 |
| minVarianceProp | Minimum ratio of the overall variance for splitting a numeric attribute. | 0.001 |
| batchSize | Number of samples to process in one batch during training. | 100 |
| numFolds | Number of folds used in cross-validation (0 means no cross-validation). | 0 |
| Seed | Seed value for random number generation, used for reproducibility of results. | 1 |
| breakTiesRandomly | Whether to randomly break ties when splitting on attributes (True/False). | False |
| Size (s) | Number of temporal partitions and corresponding random trees used in the ensemble as the size of the model. | 10 |

**Table 6 Sensitivity analysis of the TRT model across different values of the size parameter $s$ in various evaluation metrics.**

| Size (s) | Accuracy | Precision | Recall | F-measure |
|---|---|---|---|---|
| 2 | 79.01 | 0.784 | 0.790 | 0.787 |
| 3 | 82.49 | 0.812 | 0.825 | 0.819 |
| 4 | 81.86 | 0.806 | 0.819 | 0.812 |
| 5 | 82.97 | 0.816 | 0.830 | 0.823 |
| 6 | 83.20 | 0.819 | 0.832 | 0.826 |
| 7 | 83.35 | 0.821 | 0.834 | 0.827 |
| 8 | 83.47 | 0.822 | 0.835 | 0.828 |
| 9 | 83.48 | 0.823 | 0.835 | 0.829 |
| 10 | 83.54 | 0.824 | 0.835 | 0.830 |

uses the square root of the temporal index (*e.g.*, $\sqrt{1}$ to $\sqrt{s}$) to balance influence between older and newer data more moderately. We evaluated the impact of each strategy using accuracy, precision, recall, and F-measure. The results are summarized in Table 7. The proposed square-root-based voting yielded the best performance across all metrics. The simple weighted approach was followed closely, while the power-weighted scheme, despite its strong temporal bias, slightly underperformed. These results demonstrate that our square-root-based method provides a more balanced and robust integration of temporal contributions within the ensemble.

Since the datasets have temporal sequences and the chronological order of observations should be considered, the results were provided for the classifiers trained with the first 75% of the records (from November 1, 2007, to June 12, 2015) and tested with the remaining 25% of the records (from June 13, 2015, to June 25, 2017). To assess the model's effectiveness comprehensively, we utilized several standard evaluation metrics, including accuracy, precision, recall, and F-measure. The formulas for these metrics are presented in Eqs. (6) to (9), respectively. Each of these metrics provides a different perspective on the model's performance within classification tasks. The mathematical definitions for the

**Table 7 Comparison of different weighting and voting strategies in various evaluation metrics.**

| Voting strategy | Accuracy | Precision | Recall | F-measure |
|---|---|---|---|---|
| Simple weighted | 83.44 | 0.822 | 0.834 | 0.828 |
| Power-weighted | 82.96 | 0.816 | 0.830 | 0.823 |
| Square-root-based weighted | 83.54 | 0.824 | 0.835 | 0.830 |

metrics based on true positives (TP), true negatives (TN), false positives (FP), and false negatives (FN) concepts.

- Accuracy: The ratio of correctly classified instances to the total number of instances, calculated as:

$$Accuracy = \frac{TP + TN}{TP + TN + FP + FN}. \tag{6}$$

- Recall: The ratio of true positive instances correctly identified out of all actual positive instances, calculated as:

$$Recall = \frac{TP}{TP + FN}. \tag{7}$$

- Precision: The ratio of correctly predicted positive instances to the total predicted positive instances, calculated as:

$$Precision = \frac{TP}{TP + FP}. \tag{8}$$

- F-measure: It combines precision and recall metrics through their harmonic mean, ensuring a balanced evaluation of both, calculated as:

$$F\text{-}measure = 2 \times \frac{Precision \times Recall}{Precision + Recall}. \tag{9}$$

## RESULTS

The results show a clear performance improvement when using TRT compared to the standard RT model. As seen in Table 8, TRT achieved an accuracy of 83.54%, which represents a 5% improvement over RT. Furthermore, TRT outperformed RT in terms of other key metrics, including precision, recall, and F-measure. The results demonstrate that TRT, with its emphasis on recent data through its temporal weighting strategy, enhances predictive accuracy and reliability in forecasting next-day rainfall.

To further evaluate the robustness of this improvement, we conducted additional experiments using 10 different random seeds (from seed 1 to seed 10) for training both models. The goal was to assess model stability by analyzing the variance in accuracy across runs. The RT model exhibited a variance of 0.1449, while the TRT model showed a notably lower variance of 0.0692. This reduced variance indicates that TRT not only delivers higher average accuracy but also offers more consistent performance across different initializations. Such stability highlights the reliability of TRT and supports the conclusion

**Table 8 Performance comparison of RT and TRT on the WeatherAUS dataset.**

| Dataset | Accuracy (%) | | Precision | | Recall | | F-measure | |
|---|---|---|---|---|---|---|---|---|
| | RT | TRT | RT | TRT | RT | TRT | RT | TRT |
| WeatherAUS | 79.57 | 83.54 | 0.786 | 0.824 | 0.796 | 0.835 | 0.791 | 0.830 |

that the observed improvement in accuracy is not only statistically significant but also robust with respect to variations in data splits and random initialization.

The detailed performance metrics for individual locations further emphasize the effectiveness of the TRT model over the standard RT model. As presented in Table 9, which shows the accuracy of each model across numerous regional datasets, TRT consistently outperformed RT in all the regions. For instance, there is a significant increase from 74.87% with RT to 80.85% with TRT for the SydneyAirport dataset. Similarly, the accuracy was improved from 81.01% to 84.59% for the Albury dataset. For other locations, such as Cobar and Mildura, TRT also demonstrated substantial enhancements, with accuracies rising to 86.21% and 90.82%, respectively. Overall, TRT provided an average improvement of 5.1% across all datasets, reinforcing its capability to deliver more reliable predictions in rainfall classification by leveraging spatiotemporal information.

The improvements on various regional spatial datasets are not only quantitative but also address a key modeling challenge, namely the tendency of traditional models to overlook spatial heterogeneity by treating the dataset as a homogeneous whole. In contrast, TRT's evaluation was deliberately extended to individual geographic locations to capture region-specific precipitation dynamics. As detailed in Table 9, performance was assessed across 49 spatially diverse stations, where TRT consistently outperformed RT. This regional evaluation strategy serves as a practical and effective solution to the spatial integration challenge, achieving localized insight without requiring complex feature engineering or explicit geospatial modeling. The consistent accuracy gains across all regions underscore TRT's robustness in capturing local weather variations. This spatial breakdown not only validates TRT's strong generalization capability but also demonstrates its suitability for both nationwide forecasting and regionally adaptive applications.

The performance of the TRT model is further evaluated using precision, recall, and F-measure metrics, as illustrated in Fig. 5. The proposed method demonstrated a notable improvement across all three metrics compared to the standard RT method. Specifically, precision increased by 4.3%, recall improved by 5%, and F-measure showed an enhancement of 4.7% on average. These results indicate that TRT not only enhances overall accuracy but also improves the model's reliability in identifying true positive cases, which is crucial for effective rainfall prediction. By integrating spatiotemporal information, TRT successfully provided higher performance than RT across these important evaluation metrics.

The effectiveness of the TRT model can also be illustrated through the examples of confusion matrices, as shown in Table 10. For instance, in the Witchcliffe dataset, the model achieved a good accuracy of 84.82%, by accurately predicting 474 no-rain and

**Table 9 Accuracy of RT and TRT models across various regional spatial datasets.**

| Dataset | RT | TRT | Dataset | RT | TRT | Dataset | RT | TRT |
|---|---|---|---|---|---|---|---|---|
| Adelaide | 82.66 | 83.57 | Launceston | 82.03 | 82.96 | Richmond | 81.71 | 86.31 |
| Albany | 71.09 | 75.86 | Melbourne | 71.59 | 76.68 | Sale | 74.80 | 82.00 |
| Albury | 81.01 | 84.59 | MelbourneAirport | 76.49 | 81.14 | SalmonGums | 79.57 | 81.33 |
| AliceSprings | 91.56 | 92.74 | Mildura | 87.63 | 90.82 | Sydney | 74.37 | 82.04 |
| BadgerysCreek | 80.05 | 84.97 | Moree | 85.57 | 87.54 | SydneyAirport | 74.87 | 80.85 |
| Ballarat | 76.22 | 80.98 | MountGambier | 78.63 | 81.40 | Townsville | 84.98 | 88.27 |
| Bendigo | 82.21 | 85.24 | MountGinini | 73.18 | 81.29 | Tuggeranong | 80.00 | 84.67 |
| Brisbane | 81.29 | 85.59 | Newcastle | 76.59 | 77.27 | Uluru | 89.76 | 90.55 |
| Cairns | 70.28 | 77.24 | Nhil | 80.66 | 84.22 | WaggaWagga | 83.20 | 83.47 |
| Canberra | 81.05 | 84.80 | NorahHead | 75.31 | 79.13 | Walpole | 76.45 | 79.57 |
| Cobar | 82.73 | 86.21 | NorfolkIsland | 66.26 | 74.09 | Watsonia | 76.27 | 82.27 |
| CoffsHarbour | 74.70 | 80.11 | Nuriootpa | 85.22 | 86.55 | Williamtown | 72.46 | 80.44 |
| Dartmoor | 78.13 | 82.74 | PearceRAAF | 85.82 | 88.86 | Witchcliffe | 79.40 | 84.82 |
| Darwin | 81.70 | 85.21 | Penrith | 83.27 | 84.62 | Wollongong | 75.87 | 82.98 |
| GoldCoast | 72.75 | 78.39 | Perth | 85.36 | 86.73 | Woomera | 92.65 | 93.18 |
| Hobart | 76.54 | 79.80 | PerthAirport | 83.40 | 86.06 | | | |
| Katherine | 84.10 | 85.38 | Portland | 70.76 | 76.64 | | | |

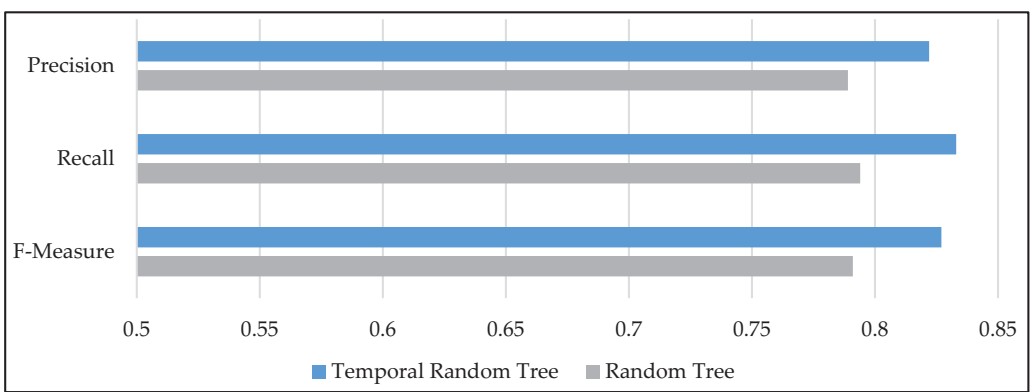

**Figure 5 The mean values of precision, recall, and F-measure for temporal random tree (TRT) and random tree (RT) models, illustrating the performance improvements of TRT.**

152 rain events, indicating a robust performance. Similarly, the model with the Dartmoor dataset demonstrated an accuracy of 82.74%, by correctly forecasting 475 no-rain and 134 rain events, reflecting its ability to effectively identify instances of rainfall. Lastly, in the MountGambier dataset, the model correctly classified 617 out of 758 instances. These example confusion matrices highlight the superior performance of the TRT model in forecasting rainfall across various locations.

Building upon the comparison of the TRT and TR models using various metrics such as accuracy, precision, recall, F-measure, and the confusion matrix, we further evaluated the

**Table 10 Confusion matrix examples for the TRT model across different datasets.**

| Dataset | Accuracy | Confusion matrix |
|---|---|---|
| Witchcliffe | 84.82% | 474 42<br>70 152 |
| Dartmoor | 82.74% | 475 48<br>79 134 |
| MountGambier | 81.40% | 466 65<br>76 151 |

statistical significance of the observed performance differences between the two models across various regional spatial datasets. To assess whether the results were statistically meaningful, we employed the Mann-Whitney U test (*Emerson, 2023*), also known as the Wilcoxon rank-sum test, which is a non-parametric test used to compare two independent samples. It tests the null hypothesis that the two groups come from the same distribution, meaning the probability of one sample being greater than the other is equal. The test is useful when the data does not meet the assumptions required for parametric tests, such as normality. It assumes that the observations are independent, and the data is at least ordinal, meaning that for any two observations, one can be ranked as greater than the other. The null hypothesis $H_0$ posits that the two groups have identical distributions, while the alternative hypothesis $H_1$ suggests that the distributions differ. This test is particularly beneficial when dealing with non-normal distributions of data. Additionally, the *p*-value obtained from this test indicates the likelihood of observing the results under the null hypothesis that the two samples come from the same distribution.

In our study, the obtained *p*-value was 0.0003422, which is far below the standard significance threshold of 0.05. According to conventional interpretations, a *p*-value less than 0.01 is considered highly significant, indicating very strong evidence against the null hypothesis. Values between 0.01 and 0.05 are deemed statistically significant, while those between 0.05 and 0.10 are often regarded as marginally significant or suggestive but not conclusive. *P*-values greater than 0.10 are typically considered not significant. Therefore, our result falls in the "highly significant" category, reinforcing the rejection of the null hypothesis and confirming that the difference between TRT and RT models is statistically meaningful.

The mathematical expression of the Mann-Whitney U test is presented in Eq. (10) as follows:

$$U = n_1 n_2 + \frac{n_1(n_1 + 1)}{2} - R_1. \tag{10}$$

In this equation, $n_1$ and $n_2$ indicate the sample sizes of the two groups being compared, while $R_1$ denotes the sum of the ranks assigned to the values in the first group. The formula uses these values to calculate the $U$ statistic, which is a measure of the difference in ranks between the two groups.

To provide a deeper understanding of the TRT method, we exemplify a part of its decision tree structure for the WeatherAUS dataset in Fig. 6. This decision tree captures

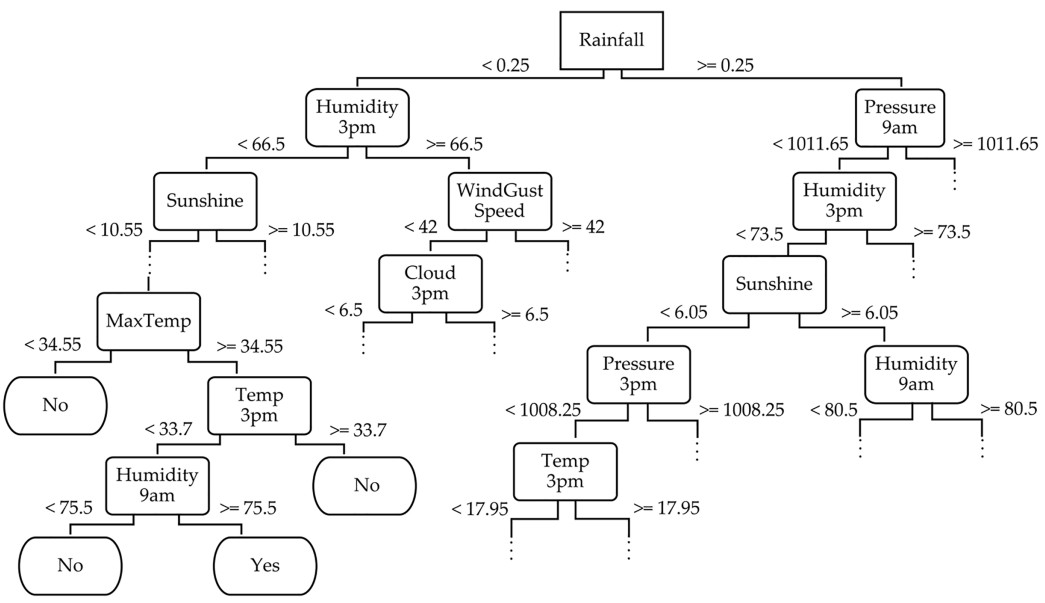

**Figure 6 Partial illustration of the temporal random tree classifier across the WeatherAUS dataset.**

the hierarchical decision-making process, where each internal node corresponds to a feature split, and the branches represent possible outcomes based on feature values. The leaf nodes indicate the final predictions. For example, the root node of the tree splits based on the "Rainfall" feature, which represents the amount of rainfall recorded for the day, helping to determine whether it will rain tomorrow by reaching the leaf nodes. If "Rainfall" <0.25, the tree further splits based on "Humidity3pm" (*e.g.*, less than 66.5), and if this is also true, it evaluates "Sunshine" (*e.g.*, less than 10.55). The leaf node in this branch might be labeled "No", indicating no rain tomorrow. Conversely, when "Rainfall" ≥0.25, the tree continues the evaluation based on features like "Pressure9am" (*e.g.*, greater than 1,011.65), followed by "Humidity3pm". This path might lead to a leaf node labeled "Yes", indicating rain tomorrow. This decision tree from the WeatherAUS dataset illustrates how the TRT method integrates both temporal and spatial features—such as humidity, pressure, sunshine, and temperature—to forecast precipitation.

In addition to illustrating the hierarchical structure, we analyzed feature importance based on their frequency of appearance near the top levels of the tree and their respective scores. Features such as "Rainfall", "Humidity3pm", and "Pressure9am" consistently appear in early splits, indicating their dominant role in shaping the prediction. This quantitative insight confirms that the model prioritizes meteorologically relevant variables, further reinforcing the success of TRT in using informative features for accurate rainfall classification.

## DISCUSSION

In this study, we compared the proposed TRT method with the state-of-the-art methods (*Akram et al., 2024*; *Asadulloh et al., 2023*; *Pangesti, Resti & Cahyono, 2023*;

**Table 11 Comparison of the TRT method with the state-of-the-art methods for rainfall classification over the WeatherAUS dataset.**

| Reference | Year | Method | Accuracy (%) | Relative performance improvement |
|---|---|---|---|---|
| *Akram et al. (2024)* | 2024 | Naive Bayes | 80.30 | 4.03 |
| | | LogitBoost | 82.20 | 1.63 |
| | | Repeated Incremental Pruning to Produce Error Reduction (RIPPER) | 82.00 | 1.88 |
| | | Decision Stump | 77.60 | 7.65 |
| | | Adaptive Boosting (AdaBoost) | 80.20 | 4.16 |
| | | Random Forest | 83.20 | 0.41 |
| | | Artificial Neural Network | 82.50 | 1.26 |
| | | K* | 79.20 | 5.48 |
| *Asaddulloh et al. (2023)* | 2023 | Support Vector Machines (Linear) | 82.29 | 1.52 |
| | | Support Vector Machines (RBF) | 77.87 | 7.28 |
| | | Support Vector Machines (Polynomial) | 78.93 | 5.84 |
| | | Support Vector Machines (Sigmoid) | 77.79 | 7.39 |
| | | Naive Bayes (Multinomial) | 74.81 | 11.67 |
| | | Naive Bayes (Gaussian) | 78.95 | 5.81 |
| | | Naive Bayes (Bernoulli) | 79.54 | 5.03 |
| | | Logistic Regression | 72.45 | 15.31 |
| *Pangesti, Resti & Cahyono (2023)* | 2023 | Fuzzy Naive Bayes | 78.19 | 6.84 |
| | | Naive Bayes | 77.13 | 8.31 |
| | | Decision Tree | 76.66 | 8.97 |
| | | Ensemble Method | 79.29 | 5.36 |
| *Mahadware et al. (2022)* | 2022 | Categorical Boosting (CatBoost) | 81.37 | 2.67 |
| | | Perceptron | 77.60 | 7.65 |
| *Umamaheswari & Ramaswamy (2022)* | 2022 | K-Nearest Neighbors (KNN) | 71.09 | 17.51 |
| | | Back Propagation Neural Network (BPNN) | 71.29 | 17.18 |
| | | Convolutional Neural Network (CNN) | 73.93 | 13.00 |
| | | Iterative Convolutional Neural Network (ICNN) | 76.78 | 8.80 |
| | | Deep Convolutional Neural Network (DCNN) | 79.85 | 4.62 |
| *He (2021)* | 2021 | Logistic Regression with Active Learning | 82.00 | 1.88 |
| | | Logistic Regression with Random Sampling Learning | 82.20 | 1.63 |
| *Zhu & Liang (2021)* | 2021 | Deep Neural Network (DNN) | 65.00 | 28.52 |
| | | Naive Bayes using Smooth Sensitivity (SSNB) | 64.00 | 30.53 |
| | | Support Vector Machines | 64.00 | 30.53 |
| | | Deep-Neural-Network-Based Model (DNN++) | 62.00 | 34.74 |
| | | Naive Bayes | 60.00 | 39.23 |
| *Siregar et al. (2020)* | 2020 | Naive Bayes | 77.22 | 8.18 |
| | | Decision Tree | 79.46 | 5.13 |
| | | Random Forest | 82.38 | 1.41 |
| Average | | | 76.52 | 9.98 |
| Proposed approach | | Temporal Random Tree (TRT) | 83.54 | |

*Mahadware et al., 2022*; *Umamaheswari & Ramaswamy, 2022*; *He, 2021*; *Zhu & Liang, 2021*; *Siregar et al., 2020*) on the same WeatherAUS dataset in Table 11 from the past five years. The results demonstrate that TRT achieved an average improvement of 9.98% compared to the existing methods. To further emphasize this improvement, we introduced the relative performance improvement column, which quantifies the percentage difference in accuracy between TRT and each baseline method. This represents how consistently TRT outperforms the others across a broad range of algorithms. By reaching the highest accuracy of 83.54%, it surpassed all other methods. This marginal improvement over previous methods suggests that TRT's ability to effectively model temporal dependencies alongside spatial information provides an edge. For example, TRT outperformed other competitive methods (*Akram et al., 2024*), such as K* (79.20%), Naive Bayes (80.30%), and RIPPER (82%), with relative performance improvement of 5.48%, 4.03%, 1.88%, respectively, reaffirming its robustness in handling complex spatiotemporal data.

Furthermore, several neural network-based approaches, including artificial neural network (ANN) (*Akram et al., 2024*), back propagation neural network (BPNN) (*Umamaheswari & Ramaswamy, 2022*), convolutional neural network (CNN), iterative CNN (ICNN), deep CNN (DCNN), and deep neural network (DNN) (*Zhu & Liang, 2021*), were evaluated. While these models generally excel in multiple prediction tasks, their performance in this study (ranging from 65.00% to 82.50%) fell short compared to TRT. A key reason lies in their limited ability to natively capture explicit temporal dependencies, especially in precipitation forecasting, where spatial and temporal interactions are key. In contrast, TRT's temporal weighting mechanism prioritizes recent data, enabling it to make more context-aware decisions based on time-sensitive meteorological patterns. This capability is particularly critical in rainfall classification, where recent weather conditions often have the greatest influence on next-day predictions. Thus, TRT's design inherently addresses a limitation often observed in conventional neural network-based methods when applied to temporally dynamic environmental data.

Methods like SVM (*Asadulloh et al., 2023*), despite their versatility across various kernels, were less effective in capturing time-dependent patterns, as seen with the lower performance of SVM (RBF) (77.87%), SVM (Polynomial) (78.93%), SVM (Linear) (82.29%), and SVM (Sigmoid) (77.79%). The proposed method clearly showed its superiority over ensemble methods (*Akram et al., 2024*) like AdaBoost (80.20%), LogitBoost (82.20%), Ensemble Method (*Pangesti, Resti & Cahyono, 2023*), CatBoost (*Mahadware et al., 2022*), and Random Forest (83.20%), with relative performance improvement of 4.16%, 1.63%, 5.36%, 2.67%, 1.41%, respectively. Although Random Forest, as a tree-based ensemble model, performed well, it still lagged behind TRT. This performance gap can be attributed to the fact that Random Forest does not incorporate a temporal weighting mechanism or prioritize recent data, whereas TRT does. The temporal focus in TRT allows it to give more weight to recent, highly informative data, which is crucial in tasks like rainfall classification, where current meteorological conditions are more relevant than older data. Decision tree-based models (*Akram et al., 2024*;

*Pangesti, Resti & Cahyono, 2023*; *Siregar et al., 2020*) such as decision stump (77.60%) showed lower accuracies compared to TRT, likely due to their weaker handling of temporal structures. In summary, the superior performance of TRT, especially over neural networks and ensemble models, underscores the advantage of its dual focus on spatial and temporal information. This makes it particularly well-suited for rainfall classification, where both dimensions are critical for accurate predictions.

## CONCLUSIONS AND FUTURE WORKS

In this work, we constructed a machine-learning-based model that can be utilized in developing proactive alert systems for timely and precise rainfall classification. This study introduces TRT, a novel approach that emphasizes the significance of recent data. Our findings indicate that TRT achieved an impressive accuracy of 83.54%, representing a 5% improvement over the standard RT model. Furthermore, TRT demonstrated enhancements across other critical evaluation measures, including precision, recall, and F-measure, highlighting its robustness and reliability in both classes (rain and no-rain). Notably, TRT consistently outperformed RT across numerous spatial datasets, showcasing significant accuracy gains in several regions. For instance, Albury's accuracy increased from 81.01% to 84.59% with TRT. Overall, TRT provided an average improvement of 5.1% across all datasets, reinforcing its capability to deliver more reliable predictions in rainfall classification through the effective integration of spatiotemporal information. This improvement was statistically significant, as confirmed by the Mann-Whitney U test with $p$-value of 0.0003422, which is well below the significance threshold of 0.05. Importantly, the proposed TRT method achieved an average improvement of 9.98% compared to the state-of-the-art techniques, with the highest accuracy of 83.54%, surpassing all other methods. Moreover, the improvements in precision (4.3%), recall (5%), and F-measure (4.7%) indicate that TRT enhances not only overall accuracy but also the model's ability to recognize true positive cases, which is essential for effective rainfall prediction. The model's design supports the development of early warning systems, enabling timely and accurate forecasts that can significantly enhance decision-making and response strategies in weather-related events.

Future work can proceed in several aspects. One promising direction is that the development of a web/mobile application can provide a user-friendly interface for accessing TRT's predictions. This application would enable real-time use of TRT's forecasting capabilities, empowering decision-makers to make timely and informed responses to weather-related events. In this way, it will be possible to make it as a versatile and important asset for precipitation prediction. In the future, in addition to rainfall prediction, the proposed TRT method can also be used in other time-varying domains such as traffic conditions, air pollution, shifting migration patterns, stock market, and remote sensing.

## APPENDIX A

Table A1 lists the abbreviations used in this study.

### Funding
The authors received no funding for this work.

### Competing Interests
The authors declare that they have no competing interests.

### Author Contributions
- Kokten Ulas Birant conceived and designed the experiments, performed the experiments, analyzed the data, prepared figures and/or tables, authored or reviewed drafts of the article, and approved the final draft.
- Bita Ghasemkhani conceived and designed the experiments, analyzed the data, performed the computation work, prepared figures and/or tables, authored or reviewed drafts of the article, and approved the final draft.
- Özlem Varlıklar conceived and designed the experiments, performed the experiments, analyzed the data, authored or reviewed drafts of the article, and approved the final draft.
- Derya Birant performed the computation work, prepared figures and/or tables, authored or reviewed drafts of the article, and approved the final draft.

### Data Availability
The Rain in Australia data is available at Kaggle: https://www.kaggle.com/datasets/jsphyg/weather-dataset-rattle-package.

### Supplemental Information
Supplemental information for this article can be found online at http://dx.doi.org/10.7717/peerj-cs.3022#supplemental-information.

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
