# Peer review of "A new machine learning method for rainfall classification: temporal random tree"

_PeerJ Computer Science, doi:10.7717/peerj-cs.3022_

## Round 0.1 · original submission · Major Revisions

· Academic Editor

Major Revisions

Major revision is required before we can consider your manuscript again.

Reviewer 1 ·

Basic reporting

In the submitted manuscript, the authors propose a new Temporal Random Tree (TRT) algorithm for forecasting and predicting precipitation. Specifically, the proposed TRT algorithm considers the temporal autocorrelation of meteorological datasets, in contrast to other Machine Learning algorithms (MLs). The proposed TRT is tested for predicting next-day (1-day lead) rain/no-rain events at 23 meteorological stations in Australia. According to the results, the proposed TRT demonstrates the most accurate predictions compared to various other MLs.

Experimental design

Overall, I believe there is value in the submitted study. However, I suggest returning this manuscript to the authors for at least one round of revision. Below, I have listed several major comments comprising my suggestions. I hope these comments will be useful to the authors in enhancing the submitted manuscript.

Major comments:
Methodology: It is suggested that the methodology section be further revised to better explain how different weights (for inputs at different time steps) are computed. I feel that some information is missing in the current manuscript. I could not find the exact input sequence length (i.e., look-back window) used for training the TRT models. Furthermore, it is unclear to me how the employed MLs are trained. More specifically, how many years of data are used for training, and how many years of data are used for validation?
Analysis and discussion: While the results of this submitted study appear promising, I also wonder what would happen if different days' meteorological data were treated as different input features. After all, tree-based ML models should, in theory, be able to identify the most sensitive and informative features through training. From this perspective, I was surprised that Random Forest actually performed worse than the proposed TRT model. I believe this is at least something worth discussing in the manuscript for the benefit of the journal's readership.

Other minor comments:
Line 195: I'm not sure if 'ACC' stands for 'accuracy.' Normally, 'ACC' is more commonly referred to as 'anomaly correlation coefficient'.
Line 227: What do you mean by 'unseen' data? Also, even for 'seen' data, shouldn't all random trees participate in the predictions?

Validity of the findings

-

Cite this review as

Reviewer 2 ·

Basic reporting

The abstract could benefit from specifying the underlying mechanism of how TRT prioritizes recent samples over older ones.

The introduction lacks a clear comparison between ML-based and traditional methods in precipitation forecasting.
Temporal dependencies in precipitation datasets should be addressed more specifically in relation to model selection.
The paper could elaborate on how TRT directly improves on other tree-based models in handling non-linearity.
Spatial data integration is mentioned, but not detailed enough in terms of algorithmic challenges and solutions.
A more in-depth discussion of how climate change impacts the variability in precipitation forecasts is needed. To improve the introduction and literature, consider the study: (A) A critical review of RNN and LSTM variants in hydrological time-series predictions. (B) Potential of artificial intelligence-based techniques for rainfall forecasting in Thailand: a comprehensive review (C) Seasonal WaveNet-LSTM: A Deep Learning Framework for Precipitation Forecasting with Integrated Large Scale Climate Drivers. (D) Advancements in daily precipitation forecasting: a deep dive into daily precipitation forecasting hybrid methods in the tropical climate of Thailand

Experimental design

The temporal weighting system could be further justified by evaluating its impact on long-term prediction accuracy.
The proposed TRT model's square-root-based weighting might require a sensitivity analysis to optimize performance across datasets.
The effect of using a weighted majority voting scheme on model robustness needs empirical validation with alternative voting strategies.
Additional experimentation with various time partitioning strategies could demonstrate the flexibility of the TRT method.
The choice of random tree classifiers needs comparison with other ensemble methods to evaluate their relative performance.
The method could be improved by integrating cross-validation to assess the stability of the TRT model across multiple runs.
A comparison with traditional machine learning models, such as decision trees or SVM, could enhance the TRT's positioning.

Validity of the findings

The claim of a 10% improvement over state-of-the-art methods needs clearer justification through detailed comparisons.
Results in Table 6 should provide more insights into performance differences across various datasets and locations.
The Mann-Whitney U test results would benefit from a confidence interval to support statistical significance.
The accuracy improvement from RT to TRT could be more thoroughly analyzed by considering data variance.
A deeper explanation of the temporal weighting strategy would enhance understanding of TRT's performance advantage.
The decision tree's hierarchical structure description lacks quantitative analysis of feature importance in prediction.
Comparison with neural network-based methods should address potential limitations in handling temporal dependencies in rainfall data.

Cite this review as

Reviewer 3 ·

Basic reporting

-

Experimental design

-

Validity of the findings

-

Additional comments

1. General Evaluation:
The manuscript introduces the Temporal Random Tree (TRT), an adaptation of the Random Tree classifier with increased weighting for recent data in precipitation classification. While the general idea—emphasizing temporality—is relevant, the manuscript lacks methodological novelty, scientific rigor, and thorough validation. The proposed method is conceptually simple, but the claims of novelty and effectiveness are overstated. Furthermore, methodological explanations are vague, empirical comparisons are insufficient, and the contextualization within the current literature, especially deep learning-based models, is weak.

2. Major Comments

2.1 Abstract
1. The abstract lacks clarity and structure—it fails to clearly define the problem, data used, study area, method, and key results (e.g., accuracy).
2. The claimed novelty is not convincing; emphasizing recent data in time-series or ensemble models is well-known. The reported 10% improvement over “state-of-the-art” is unsubstantiated due to weak and outdated comparisons.

2.2 Introduction
3. The introduction lacks proper citations—many strong claims are made without references, weakening the scientific credibility.
4. Content is generic and non-specific; it lacks scientific depth and fails to define the actual research problem or forecast horizon.
5. The terminology is inconsistent—while the title mentions “precipitation estimation,” the work focuses on rain classification; the connection to actual rainfall forecasting is unclear.
6. The practical significance of the proposed method is questionable—current numerical weather prediction models can provide reliable forecasts up to 10 days; how does this simple classifier contribute beyond that?
7. The rationale for using Random Tree (RT) instead of more robust models like Random Forest is not explained, nor are traditional forecasting methods clearly discussed.
8. Some cited references (e.g., [5]–[8]) relate to rainfall-runoff modeling, which is unrelated to rainfall classification or forecasting, and thus contextually inappropriate.
9. The contribution section is overly long but lacks scientific substance—it mainly rephrases implementation details without offering conceptual or methodological innovation.

2.3 Related Works
10. The related work section is superficial and should be merged into the introduction. Citations are listed without analysis or synthesis, and there is no critical discussion of gaps, limitations of existing models, or justification for why a new method is needed.

2.4 Methodology
11. The use of a simple weighted ensemble of Random Trees lacks novelty; if temporal data is the focus, why are state-of-the-art time-series models like LSTM not considered or compared?
12. In the Formal Description section, it is unclear whether the assigned weights for each tree are normalized—this affects interpretability and consistency of the voting mechanism.
13. Despite dealing with rainfall data, the paper omits any meteorological context—there is no mention of seasonal rainfall variability, climatic patterns, or spatiotemporal rainfall distribution, which are crucial for precipitation modeling.
14. There is no formal or visual description of the Random Tree (RT) or the proposed TRT model—readers cannot understand the structure or its differences from standard decision trees.
15. The feature selection section lacks clarity: Was the information gain used to discard features or only to rank them? Were importance values standardized across models? What was the impact of feature removal on final performance?
16. The manuscript fails to address the critical issue of class imbalance, which is inherent in binary rain/no-rain classification. As shown in the confusion matrices (e.g., Table 8), the number of no-rain samples heavily outweighs rain samples. This imbalance can severely bias the model, inflating accuracy while degrading recall for the minority class (rain). The authors do not apply any sampling strategy (e.g., oversampling, undersampling, or class weighting), nor do they report class-specific metrics like balanced accuracy or AUC. This omission casts doubt on the validity and fairness of the reported results.

2.5 Experimental Design
17. There is no description of the model tuning process—are hyperparameters fixed or optimized? The paper lacks any explanation of how parameters were selected (e.g., grid search, Bayesian optimization), raising concerns about fairness and reproducibility.
18. The manuscript does not mention whether cross-validation was used during training. Without cross-validation or robustness checks, it is unclear whether the reported performance is stable or overfitted to a specific train-test split.

2.6 References
19. The majority of references are drawn from MDPI journals and lesser-known conferences, with limited citation of top-tier, peer-reviewed journals in meteorology, hydrology, or machine learning. This lack of diversity and absence of high-impact sources undermines the scientific foundation of the study.

3. Minor Comments
• Abbreviations should be defined only once and used consistently.
• Some references are outdated or irrelevant to current ML practices.

Cite this review as

---

## Round 0.2 · accepted · Accept

· Academic Editor

Accept

Both reviewers are happy with the revision.

Reviewer 1 ·

Basic reporting

I think my previous comments are reasonably addressed.
I do not have further comments and wish the authors the best of the luck!

Experimental design

N/A

Validity of the findings

N/A

Additional comments

N/A

Cite this review as

Reviewer 2 ·

Basic reporting

Accepted

Experimental design

Accepted

Validity of the findings

Accepted

Additional comments

Accepted

Cite this review as